# r.sim.terrain 1.0: a landscape evolution model with dynamic hydrology

Brendan Alexander Harmon[1], Helena Mitasova[2,3], Anna Petrasova[2,3], and Vaclav Petras[2,3]

[1]Robert Reich School of Landscape Architecture, Louisiana State University, Baton Rouge, Louisiana, USA
[2]Center for Geospatial Analytics, North Carolina State University, Raleigh, North Carolina, USA
[3]Department of Marine, Earth, and Atmospheric Sciences, North Carolina State University, Raleigh, North Carolina, USA

**Correspondence:** Brendan Harmon (baharmon@lsu.edu)

**Abstract.** While there are numerical landscape evolution models that simulate how steady state flows of water and sediment reshape topography over long periods of time, r.sim.terrain is the first to simulate short-term topographic change for both steady state and dynamic flow regimes across a range of spatial scales. This free and open source, GIS-based topographic evolution model uses empirical models for soil erosion and a physics-based model for shallow overland water flow and soil erosion to compute short-term topographic change. This model uses either a steady state or unsteady representation of overland flow to simulate how overland sediment mass flows reshape topography for a range of hydrologic soil erosion regimes based on topographic, land cover, soil, and rainfall parameters. As demonstrated by a case study for Patterson Branch subwatershed on the Fort Bragg military installation in North Carolina, r.sim.terrain simulates the development of fine-scale morphological features including ephemeral gullies, rills, and hillslopes. Applications include land management, erosion control, landscape planning, and landscape restoration.

## 1 Introduction

Landscape evolution models represent how the surface of the earth changes over time in response to physical processes. Most studies of landscape evolution have been descriptive, but a number of numerical landscape evolution models have been developed that simulate elevational change over time (Tucker and Hancock, 2010; Temme et al., 2013). Numerical landscape evolution models such as the Geomorphic - Orogenic Landscape Evolution Model (GOLEM) (Tucker and Slingerland, 1994), CASCADE (Braun and Sambridge, 1997), the Channel-Hillslope Integrated Landscape Development (CHILD) model (Tucker et al., 2001), CAESAR (Coulthard et al., 2002, 2012), SIBERIA (Willgoose, 2005), LAPSUS (Schoorl et al., 2000, 2002) r.landscape.evol (Barton et al., 2010), and eSCAPE (Salles, 2019) simulate landscape evolution driven primarily by steady state flows over long temporal scales. Landlab, a new Python library for numerically modeling Earth surface processes (Hobley et al., 2017), has components for simulating landscape evolution such as the Stream Power with Alluvium Conservation

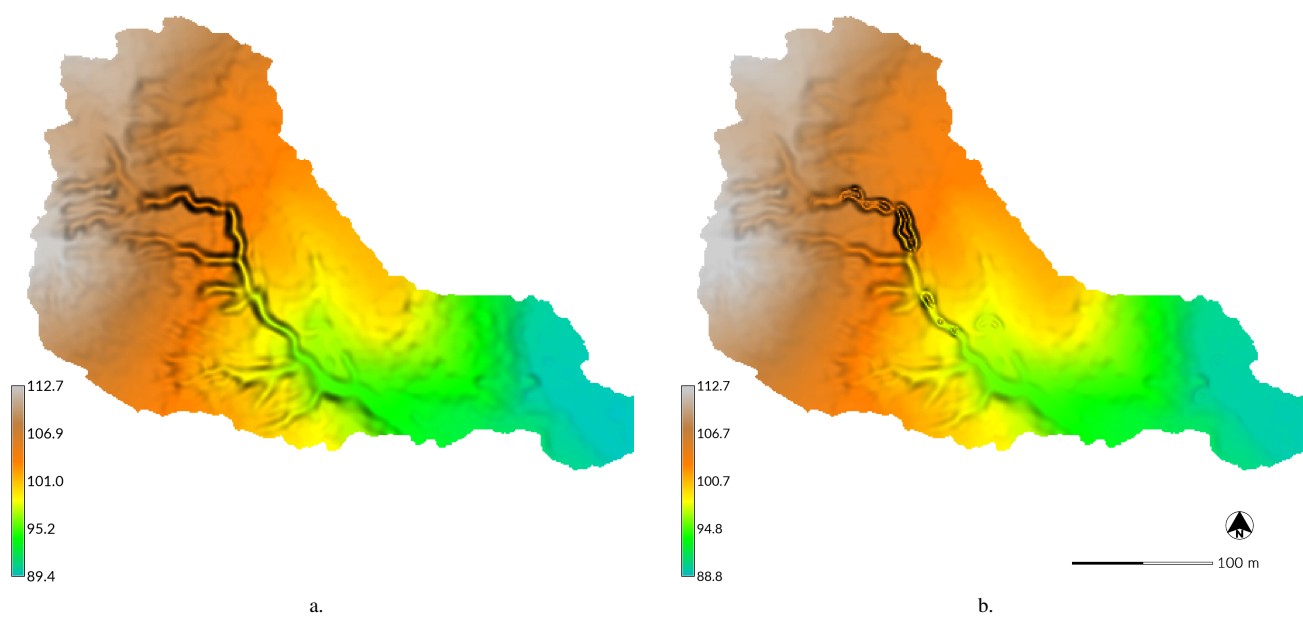

**Figure 1.** The digital elevation model (DEM) (a) before and (b) after simulated landscape evolution with r.sim.terrain for a subwatershed of Patterson Branch, Fort Bragg, NC, USA. The before DEM was generated from an airborne lidar data acquired in 2012. The simulation used the SIMWE model for a 120 min rainfall event with 50 $\text{mm hr}^{-1}$ for a variable erosion-deposition regime at steady state. In the evolved DEM the gully channel has widened with depositional ridges forming along its thalweg.

and Entrainment (SPACE) model (Shobe et al., 2017). While Geographic Information Systems (GIS) support efficient data management, spatial and statistical modeling and analysis, and visualization, there are few GIS-based soil erosion models (see Table 1) or landscape evolution models. Thaxton (2004) developed the model r.terradyn as a GRASS GIS shell script module to simulate terrain evolution by steady state net erosion-deposition rates estimated by the Simulation of Water Erosion (SIMWE) model (Mitas and Mitasova, 1998) and gravitational diffusion. Barton et al. (2010) developed a long term landscape evolution model in GRASS GIS called r.landscape.evol that integrates the Unit Stream Power Erosion Deposition (USPED) model, fluvial erosion, and gravitational diffusion. r.landscape.evol has been used to simulate the impact of prehistoric settlements on Mediterranean landscapes. In spite of the recent progress in landscape evolution modeling and monitoring, there are still major research questions to address in the theoretical foundations of erosion modeling such as how erosional processes scale over time and space and how sediment detachment and transport interact (Mitasova et al., 2013). While most numerical landscape evolution models simulate erosion processes at steady state, peak flows, short-term erosional processes like gully formation can be driven by unsteady, dynamic flow with significant morphological changes happening before flows reach steady state. A landscape evolution model with dynamic water and sediment flow is needed to study fine-scale spatial and short-term erosional processes such as gully formation and the development of microtopography.

At the beginning of a rainfall event overland water flow is unsteady – its depth changes at a variable rate over time and space. If the intensity of rainfall continues to change throughout the event then the flow regime will remain dynamic. If, however, overland flow reaches a peak rate then the hydrologic regime is considered to be at steady state. At steady state:

$$\frac{\partial h(x,y,t)}{\partial t} = 0 \qquad (1)$$

where:

$(x,y)$ is the position [m]

$t$ is the time [s]

$h(x,y,t)$ is the depth of overland flow [m].

Gullies are eroded, steep banked channels formed by ephemeral, concentrated flows of water. A gully forms when overland waterflow converges in a knickzone – a concave space with steeper slopes than its surroundings (Zahra et al., 2017) – during intense rainfall events. When the force of the water flow concentrated in the knickzone is enough to detach and transport large amounts of sediment, an incision begins to form at the apex of the knickzone – the knickpoint or headwall. As erosion continues the knickpoint begins to migrate upslope and the nascent gully channel widens, forming steep channel banks. Multiple

incisions initiated by different knickpoints may merge into a gully channel and multiple channels may merge into a branching gully system (Mitasova et al., 2013). This erosive process is dynamic; the morphological changes drive further changes in a positive feedback loop. When the gully initially forms the soil erosion regime should be detachment capacity limited with the concentrated flow of water in the channel of the gully detaching large amounts of sediment and transporting it to the foot of the gully, potentially forming a depositional fan. If the intensity of rainfall decreases and transport and detachment capacity

approach a balance, then the soil erosion regime may switch to a variable erosion-deposition regime, in which soil is eroded and deposited in a spatially variable pattern. Subsequent rainfall events may trigger further knickpoint formation and upslope migration, channel incision and widening, and depositional fan and ridge formation. Between high intensity rainfall events, lower intensity events and gravitational diffusion may gradually smooth the shape of the gully. Eventually, if detachment capacity significantly exceeds transport capacity and the regime switches to transport capacity limited, the gully may fill with

sediment as soil continues to be eroded, but can not be transported far.

Gully erosion rates and evolution can be monitored in the field or modeled on the computer. Field methods include dendrogeomorphology (Malik, 2008) and permanent monitoring stakes for recording erosion rates, extensometers for recording mass wasting events, weirs for recording water and suspended sediment discharge rates, and time series of surveys using total station theodolites (Thomas et al., 2004), unmanned aerial systems (UAS) (Jeziorska et al., 2016; Kasprak et al., 2019; Yang

et al., 2019), airborne lidar (Perroy et al., 2010; Starek et al., 2011), and terrestrial lidar (Starek et al., 2011; Bechet et al., 2016; Goodwin et al., 2016; Telling et al., 2017). With terrestrial lidar, airborne lidar, and UAS photogrammetry there is now sufficient resolution topographic data to morphometrically analyze and numerically model fine-scale landscape evolution in GIS including processes such as gully formation and the development of microtopography. Gully erosion has been simulated with RUSLE2-Raster (RUSLER) in conjunction with the Ephemeral Gully Erosion Estimator (EphGEE) (Dabney et al., 2014),

while gully evolution has been simulated for detachment capacity limited erosion regimes with the Simulation of Water Erosion (SIMWE) model (Koco, 2011; Mitasova et al., 2013). Now numerical landscape evolution models that can simulate steady state and unsteady flow regimes and can dynamically switch between soil erosion regimes are needed to study fine-scale spatial and short-term erosional processes.

5     The numerical landscape evolution model r.sim.terrain was developed to simulate the spatiotemporal evolution of landforms caused by shallow overland water and sediment flows at spatial scales ranging from square meters to kilometers and temporal scales ranging from minutes to years. This open source, GIS-based landscape evolution model can simulate either steady state or unsteady flow regimes, dynamically switch between soil erosion regimes, and simulate the evolution of fine-scale morphological features such as ephemeral gullies (Fig. 1). It was designed as a research tool for studying how erosional 10  processes scale over time and space, comparing empirical and process-based models, comparing steady state and unsteady flow regimes, and studying the role of unsteady flow regimes in fine-scale morphological change. r.sim.terrain was tested with a subwatershed scale ($450 \, \mathrm{m}^2$) case study and the simulations were compared against a time-series of airborne lidar surveys.

## 2   r.sim.terrain

The process-based, spatially distributed landscape evolution model r.sim.terrain simulates topographic changes caused by 15  shallow, overland water flow across a range of spatiotemporal scales and soil erosion regimes using either the Simulated Water Erosion (SIMWE) model, the 3-Dimensional Revised Universal Soil Loss Equation (RUSLE 3D) model, or the Unit Stream Power Erosion Deposition (USPED) model (Fig. 2). The r.sim.terrain model can simulate either steady state or dynamic flow regimes. SIMWE is a physics-based simulation that uses a Monte Carlo path sampling method to solve the water and sediment flow equations for detachment limited, transport limited, and variable erosion-deposition soil erosion regimes (Mitas and 20  Mitasova, 1998; Mitasova et al., 2004). With SIMWE r.sim.terrain uses the modeled flow of sediment – a function of water flow and soil detachment and transport parameters – to estimate net erosion and deposition rates. RUSLE3D is an empirical equation for estimating soil erosion rates in detachment capacity limited soil erosion regimes (Mitasova et al., 1996, 2013). With RUSLE3D r.sim.terrain uses an event-based rainfall erosivity factor, soil erodibility factor, landcover factor, and 3D topographic factor – a function of slope and flow accumulation – to model soil erosion rates. USPED is a semi-empirical 25  equation for net erosion and deposition in transport capacity limited soil erosion regimes (Mitasova et al., 1996, 2013). With USPED r.sim.terrain uses an event-based rainfall erosivity factor, soil erodibility factor, landcover factor, and a topographic sediment transport factor to model net erosion or deposition rates as the divergence of sediment flow. For each of the models topographic change is derived at each time step from the net erosion-deposition rate and gravitational diffusion. Depending on the input parameters, r.sim.terrain simulations with SIMWE can represent variable soil erosion-deposition regimes, including 30  prevailing detachment capacity limited or prevailing transport capacity limited regimes.

    The r.sim.terrain model can simulate the evolution of gullies including processes such as knickpoint migration, channel incision, channel widening, aggradation, scour pit formation, depositional ridge formation along the thalweg of the gully, and depositional fan formation at the foot of the gully. Applications include geomorphological research, erosion control, landscape

**r.sim.terrain**

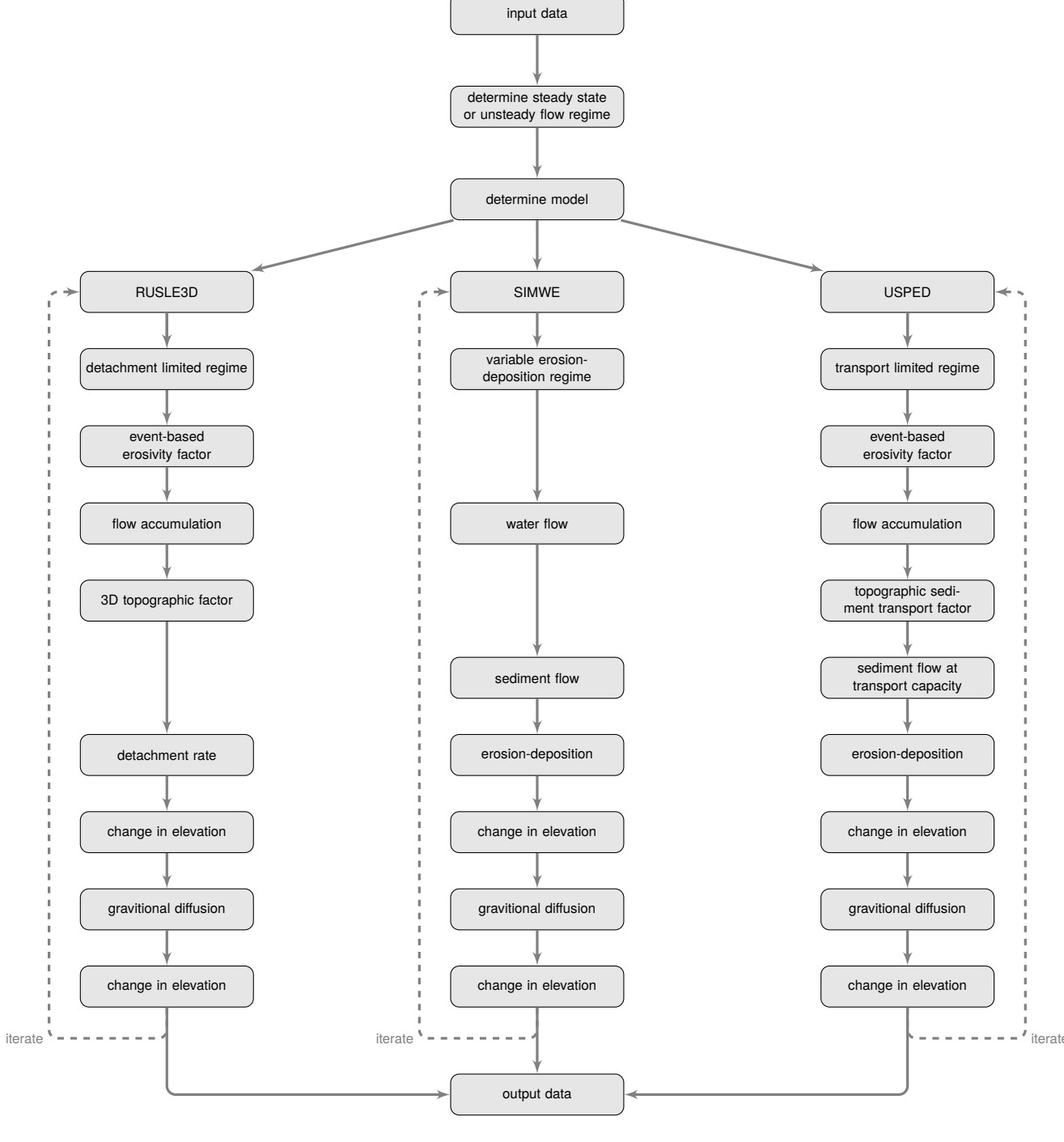

**Figure 2.** Conceptual diagram for r.sim.terrain.

**Table 1.** Examples of geospatial soil erosion models

| Model | Spatial scale | Temporal scale | Representation | Implementation | Reference |
|---|---|---|---|---|---|
| RUSLE3D | regional | continuous | raster | map algebra | (Mitasova et al., 1996) |
| USPED | watershed | continuous | raster | map algebra | (Mitasova et al., 1996) |
| SIMWE | watershed | event – continuous | raster | GRASS modules | (Mitas and Mitasova, 1998) |
| GeoWEPP | watershed | continuous | raster | ArcGIS module | (Flanagan et al., 2013) |
| AGWA | watershed | event – continuous | vector | ArcGIS module | (Guertin et al., 2015) |
| openLISEM | watershed | event | raster | PCRaster script | (Roo et al., 1996) |
| Landlab | watershed | event – continuous | raster + mesh | Python library | (Hobley et al., 2017) |

restoration, and scenario development for landscape planning and management. This model can simulate landscape evolution over a wide range of spatial scales from small watersheds less than ten square kilometers with SIMWE to regional watersheds of hundreds of square kilometers with USPED or RULSE3D, although it does not model fluvial processes. It has been used at resolutions ranging from sub-meter to 30 m. The model has been implemented as a Python add-on module for the free, open source Geographic Resources Analysis Support System (GRASS) GIS (GRASS Development Team). The source code is available at https://github.com/baharmon/landscape_evolution under the GNU General Public License v2. It supports multithreading and parallel processing to efficiently compute simulations using large, high resolution topographic datasets. The landscape evolution model can be installed in GRASS GIS as an add-on module with the command:

```
g.extension extension=r.sim.terrain
```

## 2.1 Landscape evolution

Landscape evolution in r.sim.terrain is driven by change in the elevation surface caused by soil erosion and deposition. During storm events overland flow erodes soil, transports sediment across landscape, and under favorable conditions deposits the sediment. Gravitational diffusion, applied to the changed elevation surface, simulates the smoothing effects of localized soil transport between events.

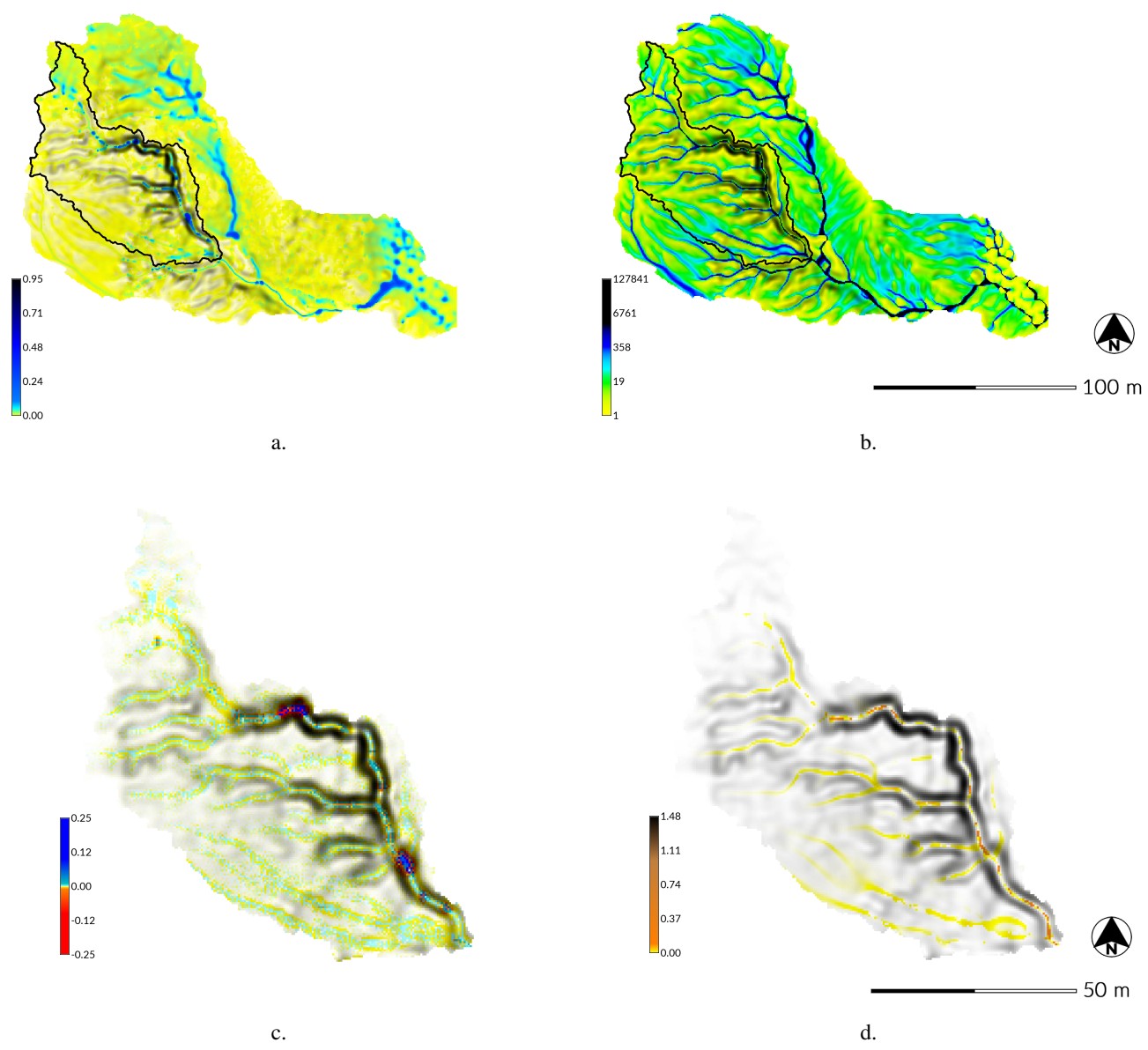

**Figure 3.** Water and sediment flows modeled with spatially variable landcover for Patterson Branch, Fort Bragg, NC: (a) water depth [m] simulated by SIMWE for a 10 min event with 50 mm hr$^{-1}$ in the subwatershed; (b) flow accumulation for RUSLE3D in the subwatershed; (c) erosion and deposition [kg m$^{-2}$ s$^{-1}$] simulated by SIMWE in drainage area 1; and (d) erosion [kg m$^{-2}$ s$^{-1}$] modeled by RUSLE3D in drainage area 1.

### 2.1.1 Elevation change

Assuming negligible uplift, the change in elevation over time is described by the continuity of mass equation expressed as the divergence of sediment flow (Tucker et al., 2001):

$$\frac{\partial z}{\partial t} = (-\nabla \cdot \boldsymbol{q_s}) \; \rho_s^{-1} = d_s \; \rho_s^{-1} \tag{2}$$

5   where:

    $z$ is elevation [m]

    $t$ is time [s]

    $\boldsymbol{q_s}$ is sediment flow per unit width (vector) [$\mathrm{kg\,m^{-1}\,s^{-1}}$]

    $d_s$ is the net erosion-deposition rate [$\mathrm{kg\,m^{-2}\,s^{-1}}$]

10     $\rho_s$ is sediment mass density [$\mathrm{kg\,m^{-3}}$].

In r.sim.terrain the net erosion-deposition rate $d_s$ driven by overland flow is estimated at different levels of complexity based on the simulation mode selected by the user. Gravitational diffusion is then applied to the changed topography to simulate the smoothing effects of localized soil transport between rainfall events. The change in elevation due to gravitational diffusion is a function of the diffusion coefficient and the Laplacian of elevation (Thaxton, 2004):

$$\frac{\partial z}{\partial t} = \varepsilon_g \; \nabla^2 z \tag{3}$$

where $\varepsilon_g$ is the diffusion coefficient [$\mathrm{m^2 s^{-1}}$].

The discrete implementation follows Thaxton (2004):

$$z_{t+\Delta t_1} = z_t + \Delta z_s \tag{4}$$

$$z_{t+\Delta t_1 + \Delta t_2} = z_{t+\Delta t_1} + \Delta z_g \tag{5}$$

where:

    $\Delta z_s$ is elevation change [m] caused by net erosion or deposition during time interval $\Delta t_1$ (Eq. 2)

    $\Delta z_g$ is the diffusion driven elevation change [m] during time interval $\Delta t_1$ (Eq. 3).

### 25   2.1.2 Erosion-deposition regimes

Following experimental observations and qualitative arguments, Foster et al. (1977) proposed that the sum of the ratio of the net erosion-deposition rate $d_s$ to the detachment capacity $D_c$ [$\mathrm{kg\,m^{-2}\,s^{-1}}$] and the ratio of the sediment flow rate $q_s = |\boldsymbol{q_s}|$ to the sediment transport capacity $T_c$ [$kg\ m^{-1}\ s^{-1}$] is a conserved quantity (unity):

$$\frac{d_s}{D_c} + \frac{q_s}{T_c} = 1 \tag{6}$$

The net erosion and deposition rate $d_s$ can then be expressed as being proportional to the difference between the sediment transport capacity $T_c$ and the actual sediment flow rate $q_s$:

$$d_s = \frac{D_c}{T_c}\left(T_c - q_s\right) \tag{7}$$

This principle is used in several erosion models including the Water Erosion Prediction Project (WEPP) (Flanagan et al., 2013) and SIMWE (Mitas and Mitasova, 1998).

Using this concept it is possible to identify two limiting erosion-deposition regimes. When $T_c \gg D_c$ leading to $T_c \gg q_s$, the erosion regime is detachment capacity limited and net erosion is equal to the detachment capacity:

$$d_s = D_c \tag{8}$$

For this case the transport capacity of overland flow exceeds the detachment capacity and thus sediment flow, erosion, and sediment transport are limited by the detachment capacity. Therefore, no deposition occurs. An example of this case is when a strong storm producing intense overland flow over compacted clay soils causes high capacity flows to transport light clay particles, while the detachment of compacted soils is limited. When $D_c \gg T_c$, sediment flow is at sediment transport capacity $q_s = T_c$, leading to a transport capacity limited regime with deposition reaching its maximum extent for the given water flow. Net erosion-deposition is computed as the divergence of transport capacity multiplied by a unit vector $s_0$ in the direction of flow:

$$d_s = \nabla \cdot (T_c\, s_0) \tag{9}$$

This case may occur, for example, during a moderate storm with overland flow over sandy soils with high detachment capacity, but low transport capacity. For $0 < (D_c/T_c) < \infty$ the spatial pattern of net erosion-deposition is variable and depends on the difference between the sediment transport capacity and the actual sediment flow rate at the given location.

The detachment capacity $D_c$ and the sediment-transport capacity $T_c$ are estimated using shear stress and stream power equations respectively expressed as power functions of water-flow properties and slope angle. The relations between the topographic parameters of well known empirical equations for erosion modeling, such as USLE and stream power, were presented by Moore and Burch (1986) and used to develop simple, GIS-based models for limiting erosion-deposition cases such as RUSLE3D and USPED (Mitasova and Mitas, 2001). The SIMWE model estimates $T_c$ and $D_c$ using modified equations and parameters developed for the WEPP model (Flanagan et al., 2013; Mitasova et al., 2013).

The simulation modes in r.sim.terrain include (Fig. 2):

- the process-based SIMWE model for steady state and unsteady shallow overland flow in variable erosion-deposition regimes with $d_s$ computed by solving the shallow water flow and sediment transport continuity equations,

- the RUSLE3D model for detachment capacity limited cases with $d_s$ given by Eq. (8),

- and the USPED model for transport capacity limited regimes with $d_s$ given by Eq. (9).

The following sections explain the computation of $d_s$ for these three modes in more detail.

## 2.2 Simulation of Water Erosion (SIMWE)

SIMWE is a physics-based simulation of shallow overland water and sediment flow that uses a path sampling method to solve the continuity equations with a 2D diffusive wave approximation (Mitas and Mitasova, 1998; Mitasova et al., 2004). SIMWE has been implemented in GRASS GIS as the modules r.sim.water and r.sim.sediment. In SIMWE mode for each landscape evolution time step r.sim.terrain:

- computes the first order partial derivatives of the elevation surface $\partial z/\partial x$ and $\partial z/\partial y$,

- simulates shallow water flow depth, sediment flow, and the net erosion-deposition rate,

- and then evolves the topography based on the erosion-deposition rate and gravitational diffusion.

The first order partial derivatives of the elevation surface are computed using the GRASS GIS module r.slope.aspect using the

10 equations in Hofierka et al. (2009). r.sim.terrain simulates unsteady state flow regimes when the landscape evolution time step is less than the travel time for a drop of water or a particle of sediment to cross the landscape, e.g. when the time step is less than the time to concentration for the modeled watershed. With longer landscape evolution time steps the model simulates a steady state regime.

### 2.2.1 Shallow water flow

The SIMWE model simulates shallow overland water flow controlled by spatially variable topographic, soil, landcover, and rainfall parameters using a Green's function Monte Carlo path sampling method. The steady state shallow water flow continuity equation relates the change in water depth across space to source, defined in our case as rainfall excess rate:

$$\nabla \cdot \boldsymbol{q} = \nabla \cdot (h\,\boldsymbol{v}) = \nabla \cdot (n^{-1}\,h^{5/3}s^{1/2}\boldsymbol{s_0}) = i_e \tag{10}$$

where:

$\boldsymbol{q}$ is the water flow per unit width (vector) $[\mathrm{m}^2\,\mathrm{s}^{-1}]$

$h$ is the depth of overland flow $[\mathrm{m}]$

$\boldsymbol{v}$ is the water flow velocity vector $[\mathrm{m}\,\mathrm{s}^{-1}]$ whose magnitude is computed with the Manning's equation $v = n^{-1}\,h^{2/3}\,s^{1/2}$

$n$ is the Manning's coefficient $[\mathrm{s}\,\mathrm{m}^{-1/3}]$

$s$ is slope (unitless)

$i_e$ is the rainfall excess rate $[\mathrm{m}\,\mathrm{s}^{-1}]$ (i.e. rainfall intensity − infiltration − vegetation intercept).

An approximation of diffusive wave effects is incorporated by adding a diffusion term proportional to $\nabla^2[h^{5/3}]$:

$$-\frac{\varepsilon_w}{2}\nabla^2\,h^{5/3} + \nabla \cdot (n^{-1}\,h^{5/3}s^{1/2}\boldsymbol{s_0}) = i_e \tag{11}$$

where:

$\varepsilon_w$ is a spatially variable diffusion coefficient $[\mathrm{m}^{4/3}\,\mathrm{s}^{-1}]$.

The path sampling method solves the continuity equation for $h^{5/3}$ through the accumulation of the evolving source (Mitasova et al., 2004). The solution assumes that water flow velocity is largely controlled by the slope of the terrain and surface roughness and that its change at a given location during the simulated event is negligible. The initial number of particles per grid cell is proportional to the rainfall excess rate $i_e$ (source). The water depth $h^{5/3}$ at time $\tau$ during the simulated rainfall event is computed as a function of particle (walkers) density at each grid cell. Particles are routed across the landscape by finding a new position for each walker at time $\tau + \Delta\tau$:

$$\boldsymbol{r}_m^{new} = \boldsymbol{r}_m + \Delta\tau\boldsymbol{v} + \boldsymbol{g} \tag{12}$$

where:

$\boldsymbol{r} = (x, y)$ is the $m^{th}$ walker position [m]

$\Delta\tau$ is the particle routing time step [s]

$\boldsymbol{g}$ is a random vector with gaussian components with variance $\Delta\tau$ [m].

The mathematical background of the method, including the computation of the temporal evolution of water depth and incorporation of approximate momentum through an increased diffusion rate in the prevailing direction of flow, is presented by Mitas and Mitasova (1998) and Mitasova et al. (2004).

### 2.2.2 Sediment flow and net erosion-deposition

The SIMWE model simulates the sediment flow over complex topography with spatially variable overland flow, soil, and landcover properties by solving the sediment flow continuity equation using a Green's function Monte Carlo path sampling method. Steady state sediment flow $\boldsymbol{q_s}$ is approximated by the bivariate continuity equation, which relates the change in sediment flow rate to effective sources and sinks:

$$\nabla \cdot \boldsymbol{q_s} = \text{sources} - \text{sinks} = d_s \tag{13}$$

The sediment-flow rate $\boldsymbol{q_s}$ is a function of water flow and sediment concentration (Mitas and Mitasova, 1998):

$$\boldsymbol{q_s} = \rho_s\, c\, \boldsymbol{q} = \rho_s\, c\, h\, \boldsymbol{v} = \varrho\, \boldsymbol{v} \tag{14}$$

where:

$\rho_s$ is sediment mass density in the water column [$\text{kg m}^{-3}$]

$c$ is sediment concentration [$\text{particle m}^{-3}$]

$\varrho = \rho_s\, c\, h$ is the mass of sediment transported by water per unit area [$\text{kg m}^{-2}$].

The sediment flow equation (Eq. 13), like the water flow equation, has been rewritten to include a small diffusion term that is proportional to the mass of water-carried sediment per unit area $\nabla^2\varrho$ (Mitas and Mitasova, 1998):

$$-\frac{\varepsilon_s}{2}\nabla^2\varrho + \nabla \cdot (\varrho\boldsymbol{v}) + \varrho\,\frac{D_c}{T_c}\,|\boldsymbol{v}| = d_s \tag{15}$$

where:

$\varepsilon_s$ is the diffusion constant $[\mathrm{m}^2\,\mathrm{s}^{-1}]$.

On the left hand side of Eq. (15) the first term describes local diffusion, the second term is drift driven by water flow, and the third term represents a velocity dependent 'potential' acting on the mass of transported sediment. The initial number of particles per grid cell is proportional to the soil detachment capacity $D_c$ (source). The particles are then routed across the landscape by finding a new position for each walker at time $\tau + \Delta\tau$:

$$\boldsymbol{r}_m^{new} = \boldsymbol{r}_m + \Delta\tau\boldsymbol{v} + \boldsymbol{g} \tag{16}$$

while the updated weight is:

$$w_m^{new} = w_m \exp[-\Delta\tau(u(\boldsymbol{r}_m^{new}) + u(\boldsymbol{r}_m))/2] \tag{17}$$

where:

$u = (D_c/T_c)\,|\boldsymbol{v}|$.

Sediment flow is computed as the product of weighted particle densities and the water flow velocity (Eq. 14) and the net erosion-deposition rate $d_s$ is computed as the divergence of sediment flow using Eq. (13). See Mitas and Mitasova (1998) and Mitasova et al. (2004) for more details on the Green's function Monte Carlo solution and equations for computing $D_c$ and $T_c$.

This model can simulate erosion regimes from prevailing detachment limited conditions when $T_c >> D_c$ to prevailing transport capacity limited conditions when $D_c >> T_c$ and the erosion-deposition patterns between these conditions. At each landscape evolution time step, the regime can change based on the ratio between the sediment detachment capacity $D_c$ and the sediment transport capacity $T_c$ and the actual sediment flow rate. If the landscape evolution time step is shorter than the time to concentration (i.e. the time for water to reach steady state) then net erosion-deposition is derived from unsteady flow.

## 2.3 Revised Universal Soil Loss Equation for Complex Terrain (RUSLE3D)

RUSLE3D is an empirical model for computing erosion in a detachment capacity limited soil erosion regime for watersheds with complex topography (Mitasova et al., 1996). It is based on the Universal Soil Loss Equation (USLE), an empirical equation for estimating the average sheet and rill soil erosion from rainfall and runoff on agricultural fields and rangelands with simple topography (Wischmeier et al., 1978). It models erosion dominated regimes without deposition in which sediment transport capacity is uniformly greater than detachment capacity. In USLE soil loss per unit area is determined by an erosivity factor $R$, a soil erodibility factor $K$, a slope length factor $L$, a slope steepness factor $S$, a cover management factor $C$, and a prevention measures factor $P$. These factors are empirical constants derived from an extensive collection of measurements on 22.13 m standard plots with an average slope of 9%. RUSLE3D was designed to account for more complex, 3D topography with converging and diverging flows. In RUSLE3D the topographic potential for erosion at any given point is represented by a 3D topographic factor $LS_{3D}$, which is a function of the upslope contributing area and the angle of the slope.

In this spatially and temporally distributed model RUSLE3D is modified by the use of a event-based R-factor derived from rainfall intensity at each time step. For each time step this model computes the parameters for RUSLE3D – an event-

based erosivity factor, the slope of the topography, the flow accumulation, and the 3D topographic factor – and then solves the RUSLE3D equation for the rate of soil loss (i.e. the net soil erosion rate). The soil erosion rate is then used to simulate landscape evolution in a detachment capacity limited soil erosion regime.

### 2.3.1 Erosivity factor

The erosivity factor $R$ in USLE and RUSLE is the combination of the total energy and peak intensity of a rainfall event, representing the interaction between the detachment of sediment particles and the transport capacity of the flow. It can be calculated as the product of the the kinetic energy of the rainfall event $E$ and its maximum 30 min intensity $I_{30}$ (Brown and Foster, 1987; Renard et al., 1997; Panagos et al., 2015, 2017). In this model, however, the erosivity factor is derived at each time step as a function of kinetic energy, rainfall depth, rainfall intensity, and time. First rain energy is derived from rainfall
intensity (Brown and Foster, 1987; Yin et al., 2017):

$$\frac{e_r}{e_0} = 1. - b \, exp\left(\frac{i_r}{i_0}\right) \tag{18}$$

where:

$e_r$ is unit rain energy [MJ ha$^{-1}$ mm$^-$1]

$i_r$ is rainfall intensity [mm h$^{-1}$]

$b$ is empirical coeficient

$i_0$ is reference rainfall intensity [mm h$^{-1}$]

$e_0$ is reference energy [MJ ha$^{-1}$ mm$^-$1].

The parameters for this equation were derived from observed data published for different regions by Panagos et al. (2017). Then the event-based erosivity index $R_e$ is calculated as the product of unit rain energy, rainfall depth, rainfall intensity, and
time:

$$R_e = e_r \, v_r \, i_r \, \Delta t \tag{19}$$

where:

$R_e$ is the event-based erosivity index [MJ mm ha$^{-1}$ hr$^{-1}$]

$v_r$ is the rainfall depth [mm] derived from $v_r = i_r \, \Delta t$

$\Delta t$ is the change in time [s].

### 2.3.2 Flow accumulation

The upslope contributing area per unit width $a$ is determined by flow accumulation (the number of grid cells draining into a given grid cell) multiplied by grid cell width (Fig. 3d). Flow accumulation is calculated using a multiple flow direction algorithm (Metz et al., 2009) based on $A^T$ least cost path searches (Ehlschlaeger, 1989). The multiple flow direction algorithm

implemented in GRASS GIS as the module r.watershed is computationally efficient, does not require sink filling, and can navigate nested depressions and other obstacles.

### 2.3.3 3D topographic factor

The 3D topographic factor $LS_{3D}$ is calculated as a function of the upslope contributing area and the slope (Fig. 3e).

$$LS_{3D} = (m+1) \left( \frac{a}{a_0} \right)^m \left( \frac{\sin\beta}{\beta_0} \right)^n \tag{20}$$

where:

$LS_{3D}$ is the dimensionless topographic factor

$a$ is upslope contributing area per unit width [m]

$a_0$ is the length of the standard USLE plot [22.1 m]

$\beta$ is the angle of the slope [°]

$m$ is an empirical coefficient

$n$ is an empirical coefficient

$\beta_0$ is the slope of the standard USLE plot [5.14°].

The empirical coefficients $m$ and $n$ for the upslope contributing area and the slope can range from 0.2 to 0.6 and 1.0 to 1.3 respectively with low values representing dominant sheet flow and high values representing dominant rill flow.

### 2.3.4 Detachment limited erosion rate

The erosion rate is a function of the event-based erosivity factor, soil erodibility factor, 3D topographic factor, landcover factor, and prevention measures factor (Fig. 3d):

$$E = R_e \ K \ LS_{3D} \ C \ P \tag{21}$$

where:

$E$ is soil erosion rate (soil loss) [$\mathrm{kg\,m^{-2}\,min^{-1}}$]

$R_e$ is the event-based erosivity factor [$\mathrm{MJ\,mm\,ha^{-1}\,hr^{-1}}$]

$K$ is the soil erodibility factor [$\mathrm{ton\,ha\,hr\,ha^{-1}\,MJ^{-1}\,mm^{-1}}$]

$LS_{3D}$ is the dimensionless topographic (length-slope) factor

$C$ is the dimensionless landcover factor

$P$ is the dimensionless prevention measures factor.

The detachment limited erosion represented by RUSLE3D leads to the simulated change in elevation:

$$\Delta z_s = D_c \rho_s^{-1} = E \ \rho_s^{-1} \tag{22}$$

which is combined with Eq. (3) for gravitational diffusion.

### 2.4 Unit Streampower Erosion Deposition (USPED)

USPED estimates net erosion-deposition as the divergence of sediment flow in a transport capacity limited soil erosion regime. The amount of soil detached is close to the amount of sediment that water flow can carry. As a transport capacity limited model USPED predicts erosion where transport capacity increases and deposition where transport capacity decreases. The influence of topography on sediment flow is represented by a topographic sediment transport factor, while the influence of soil and landcover are represented by factors adopted from USLE and RUSLE (Mitasova et al., 1996). Sediment flow is estimated by computing the event-based erosivity factor ($R_e$) using Eq. (19), the slope and aspect of the topography, the flow accumulation with a multiple flow direction algorithm, the topographic sediment transport factor, and sediment flow at transport capacity. Net erosion-deposition is then computed as the divergence of sediment flow.

### 2.4.1 Topographic sediment transport factor

Using the unit stream power concept presented by Moore and Burch (1986), the 3D topographic factor (Eq. 20) for RUSLE3D is modified to represent the topographic sediment transport factor ($LS_T$) – the topographic component of overland flow at sediment transport capacity:

$$LS_T = a^m \, (\sin \beta)^n \qquad (23)$$

where:

    $LS_T$ is the topographic sediment transport factor

    $a$ is the upslope contributing area per unit width [m]

    $\beta$ is the angle of the slope [°]

    $m$ is an empirical coefficient

    $n$ is an empirical coefficient.

### 2.4.2 Transport limited sediment flow and net erosion-deposition

Sediment flow at transport capacity is a function of the event-based rainfall factor, soil erodibility factor, topographic sediment transport factor, landcover factor, and prevention measures factor:

$$T = R_e \, K \, C \, P \, LS_T \qquad (24)$$

where:

    $T$ is sediment flow at transport capacity [$\mathrm{kg \, m^{-1} \, s^{-1}}$]

    $R_e$ is the event-based rainfall factor [$\mathrm{MJ \, mm \, ha^{-1} \, hr^{-1}}$]

    $K$ is the soil erodibility factor [$\mathrm{ton \, ha \, hr \, ha^{-1} \, MJ^{-1} \, mm^{-1}}$]

    $C$ is the dimensionless land cover factor

$P$ is the dimensionless prevention measures factor.

Net erosion-deposition is estimated as the divergence of sediment flow, assuming that sediment flow is equal to sediment transport capacity:

$$d_s = \frac{\partial(T_c \cos\alpha)}{\partial x} + \frac{\partial(T_c \sin\alpha)}{\partial y} \tag{25}$$

where:

$d_s$ is net erosion-deposition $[\mathrm{kg\,m^{-2}\,s^{-1}}]$

$\alpha$ is the aspect of the topography (i.e. the direction of flow) $[°]$.

With USPED the simulated change in elevation $\Delta z_s = d_s$ is derived from Eq. (2) for landscape evolution and then Eq. (3) for gravitational diffusion.

## 3  Case study

Military activity is a high-impact land use that can cause significant physical alteration to the landscape. Erosion is a major concern for military installations, particularly at training bases, where the land surface is disturbed by off-road vehicles, foot traffic, and munitions. Off-road vehicles and foot traffic by soldiers cause the loss of vegetative cover, the disruption of soil structure, soil compaction, and increased runoff due to reduced soil capacity for water infiltration (Webb and Wilshire, 1983; McDonald, 2004). Gullies – ephemeral channels with steep headwalls that incise into unconsolidated soil to depths of meters – are a manifestation of erosion common to military training installations like Ft. Bragg in North Carolina and the Piñon Canyon Maneuver Site in Colorado. While the local development of gullies can restrict the maneuverability of troops and vehicles during training exercises, pervasive gullying across a landscape can degrade an entire training area (Huang and Niemann, 2014). To test the effectiveness of the different models in r.sim.terrain we compared the simulated evolution of a highly eroded subwatershed of Patterson Branch on Fort Bragg, North Carolina against a timeseries of airborne lidar surveys. The models – SIMWE, RUSLE3D, and USPED – were tested in steady state and dynamic modes for design storms with constant rainfall.

### 3.1  Patterson Branch

With $650 \ \mathrm{km^2}$ of land Fort Bragg is the largest military installation in the US and has extensive areas of bare, erodible soils on impact areas, firing ranges, landing zones, and dropzones. It is located in the Sandhills region of North Carolina with a Longleaf Pine and Wiregrass Ecosystem (Sorrie et al., 2006). The study landscape – a subwatershed of Patterson Branch (Fig. 4) in the Coleman Impact Area – is pitted with impact craters from artillery and mortar shells and has an active, approximately $2 \ \mathrm{m}$ deep gully. It is a Pine-Scrub Oak Sandhill community composed primarily of Longleaf Pine (*Pinus palustris*) and Wiregrass (*Aristida stricta*) on Blaney and Gilead loamy sands (Sorrie, 2004). Throughout the Coleman Impact Area frequent fires ignited by live munitions drive the ecological disturbance regime of this fire adapted ecosystem. In 2016 the $450 \ \mathrm{m^2}$ study site was

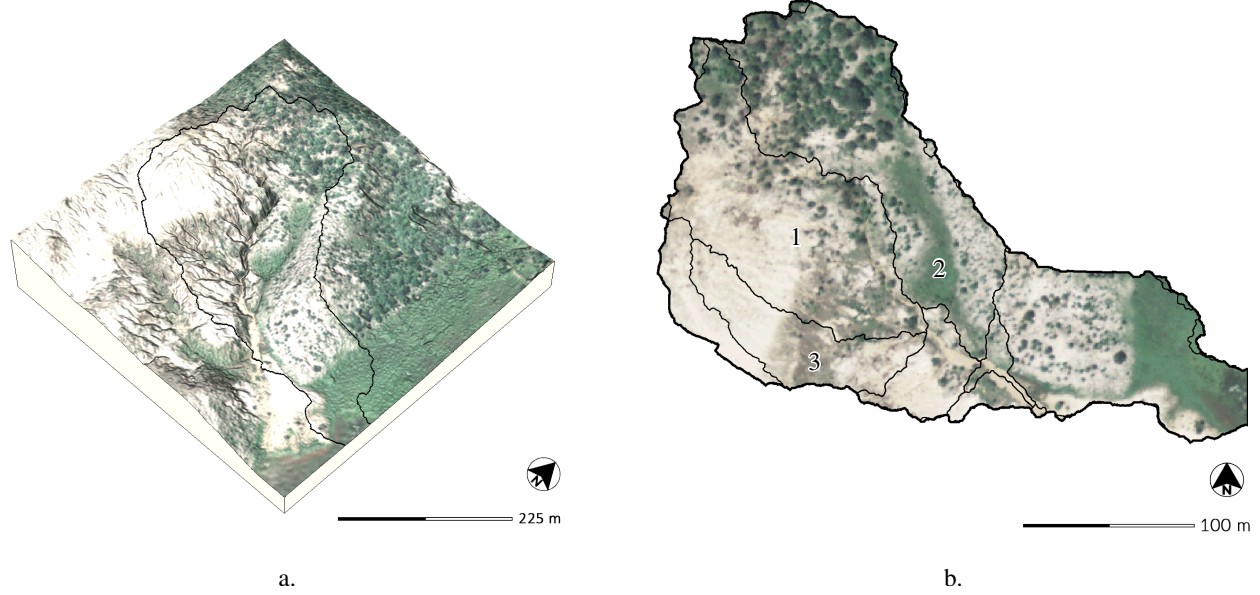

a.                                                    b.

**Figure 4.** Subwatershed with 2014 orthoimagery (a) draped over the 2016 digital elevation model and (b) drainage areas with 2014 orthoimagery, Patterson Branch, Fort Bragg, NC, USA.

43.24% bare ground with predominately loamy sands, 39.54% covered by the Wiregrass community, and 17.22% forested with the Longleaf Pine community (Fig. 5a). We hypothesize that the elimination of forest cover in the impact zone triggered extensive channelized overland flow, gully formation, and sediment transport into the creek.

Timeseries of digital elevations models and landcover maps for the study landscape were generated from lidar pointclouds and orthophotography. The digital elevations models for 2004, 2012, and 2016 were interpolated at 1 m resolution using the regularized spline with tension function (Mitasova and Mitas, 1993; Mitasova et al., 2005) from airborne lidar surveys collected by the NC Floodplain Mapping program and Fort Bragg. Unsupervised image classification was used to identify clusters of spectral reflectance in a timeseries of 1 m resolution orthoimagery collected by the National Agriculture Imagery Program. The landcover maps were derived from the classified lidar point clouds and the classified orthoimagery. Spatially variable soil erosion factors – the k-factor, c-factor, Manning's coefficient, and runoff rate – were then derived from the landcover and soil maps. The dataset for this study is hosted at https://github.com/baharmon/landscape_evolution_dataset under the ODC Open Database License (ODbL). The data is derived from publicly available data from the US Army, USGS, USDA, Wake County GIS, NC Floodplain Mapping Program, and the NC State Climate Office. There are detailed instructions for preparing the input data in the tutorial and a complete record of the commands used to process the sample data in the data log.

We used the geomorphons method of automated landform classification based on the openness of terrain (Jasiewicz and Stepinski, 2013) and the difference between the digital elevation models to analyze the changing morphology of the study

area (Fig. 5 & 6). The 2 m deep gully – its channels classified as valleys and its scour pits as depressions by geomorphons – has multiple mature branches and ends with a depositional fan. The gully has also developed depositional ridges beside the channels. Deep scour pits have developed where branches join the main channel and where the main channel has sharp bends. A new branch has begun to form in a knickzone classified as a mix of valleys and hollows on a grassy swale on the northeast side of the gully. Between 2012 and 2016 a depositional ridge developed at the foot of this nascent branch where it would meet the main channel. The 2016 minus 2012 DEM of Difference (DoD) – i.e. the difference in elevation (Fig. 5c & 6c) – shows a deepening of the main channel by approximately 0.2 m and scours pits by approximately 1 m, while depositional ridges have formed and grown up to approximately 1 m high. The DoD also shows that 244.60 m$^3$ of sediment were deposited on the depositional fan between 2012 and 2016.

## 3.2   Simulations

We ran a sequence of r.sim.terrain simulations with design storms for the Patterson Branch subwatershed study area to demonstrate the capabilities of the RUSLE3D, USPED, and SIMWE models (Table 2). To analyze the results of the simulations, we compared net differences in elevation morphological features, and volumetric change. While r.sim.terrain can use rainfall records, we used design storms to demonstrate and test the basic capabilities of the model. Our design storms were based off the peak rainfall values in records from the State Climate Office of North Carolina. We used RUSLE3D to simulate landscape evolution in a dynamic, detachment capacity limited soil erosion regime for a 120 min design storm with 3 min intervals and a constant rainfall intensity of 50 mm hr$^{-1}$ (Fig. 7). We used USPED to simulate landscape evolution in a dynamic, transport capacity limited soil erosion regime for a 120 min design storm with 3 min intervals and a constant rainfall intensity of 50 mm hr$^{-1}$ (Fig. 8). We used SIMWE to simulate landscape evolution in a steady state, variable erosion-deposition soil erosion regime for a 120 min design storm with a constant rainfall intensity of 50 mm hr$^{-1}$ (Fig. 9). In all of the simulations a sink filling algorithm – an optional parameter in r.sim.terrain – was used to reduce the effects of positive feedback loops that cause the over-development of scour pits.

The simulations were automated and run in parallel using Python scripts that are available in the software repository. The simulations can be reproduced using these scripts and the study area dataset by following the instructions in the Open Science Framework repository at https://osf.io/tf6yb/. The simulations were run in GRASS GIS 7.4 on a desktop computer with 64-bit Ubuntu 16.04.4 LTS, 8 x 4.20 GHz Intel Core i7 7700K CPUs, and 32 GB RAM. Simulations using SIMWE are far more computationally intensive than RULSE3D or USPED, but support multi-threading when compiled with OpenMP. Dynamic simulations of RUSLE3D and USPED took 2 min 36 s and 3 min 14 s respectively to run on a single thread, while the steady state simulation for SIMWE took 44 min 51 s running on 6 threads (Table 2).

## 3.3   Results

We used the Difference in DEMs to compute volumetric changes between the lidar surveys and the simulations (Table 3). We applied a threshold of ±0.18 m to the lidar surveys since they had a vertical accuracy at a 95% confidence level of 18.15 cm based on a 9.25 cm root mean square error (RMSEz) for non vegetated areas in accordance with the National Digital Elevation

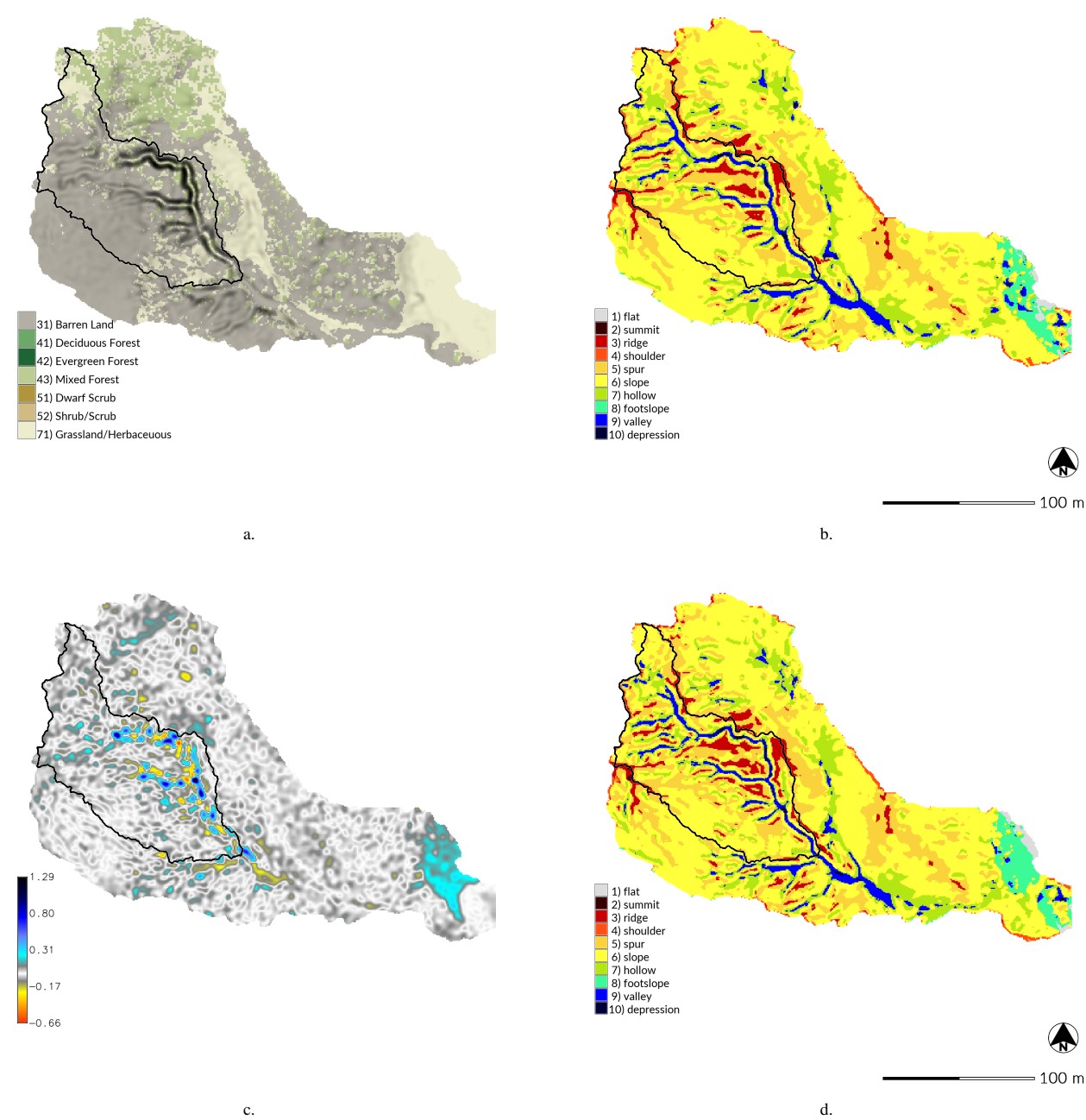

**Figure 5.** Morphological change in the subwatershed of Patterson Branch, Fort Bragg, NC, USA: (a) landcover in 2014, (b) landforms in 2012, (c) elevation difference between 2012-2016 [m], and (d) landforms in 2016.

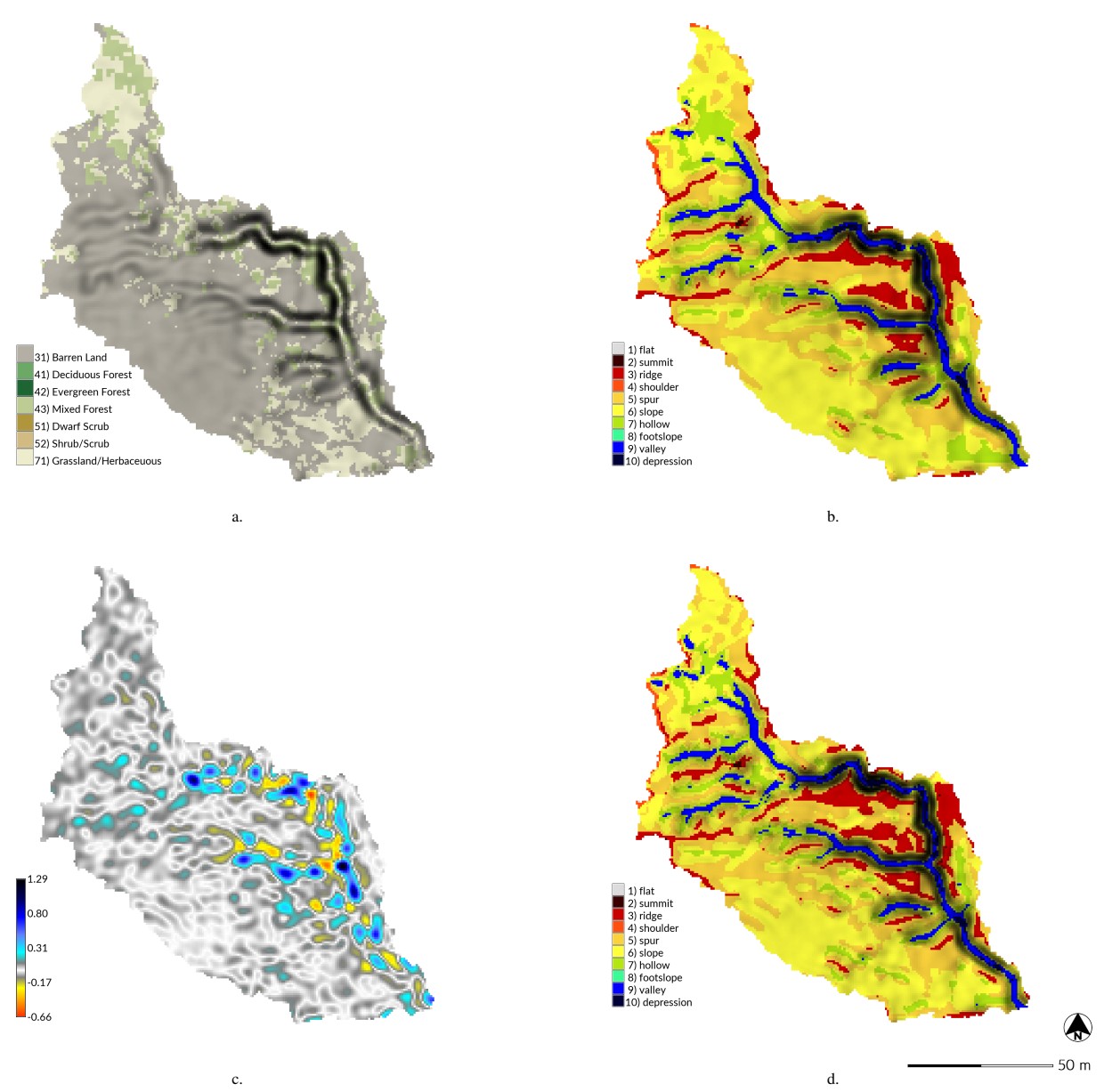

**Figure 6.** Detailed morphological change for drainage area 1 of Patterson Branch, Fort Bragg, NC, USA: (a) landcover in 2014, (b) landforms in 2012, (c) elevation difference between 2012-2016 [m], and (d) landforms in 2016.

**Table 2.** Landscape evolution simulations

| Flow regime | Model | Intensity | Duration | Interval | m | n | $\rho_s$ | Threads | Runtime |
|---|---|---|---|---|---|---|---|---|---|
| Dynamic | RUSLE3D | 50 mm hr$^{-1}$ | 120 min | 3 min | 0.4 | 1.3 | | | 2 min 36 s |
| Dynamic | USPED | 50 mm hr$^{-1}$ | 120 min | 3 min | 1.5 | 1.2 | 1.6 | | 3 min 14 s |
| Steady state | SIMWE | 50 mm hr$^{-1}$ | 120 min | 120 min | | | 1.6 | 6 | 44 min 51 s |

**Table 3.** Volumetric change

| Difference of DEMs (DoD) | Threshold [m] | Erosion [m$^3$] | Deposition [m$^3$] | Net change [m$^3$] |
|---|---|---|---|---|
| 2016 - 2012 | ±0.18 | 152.96 | 807.74 | 654.77 |
| Simulated with RUSLE3D - 2012 | None | 1480.75 | 0 | -1480.75 |
| Simulated with USPED - 2012 | None | 1235.08 | 727.46 | -507.62 |
| Simulated with SIMWE -2012 | None | 758.56 | 608.91 | -149.66 |

Program guidelines (North Carolina Risk Management Office, 2018). Given the presence of the mature gully with ridges along its banks, we hypothesize that the study landscape had previously been dominated by a detachment limited soil erosion regime, but – given the net change of 654.77 m$^3$ – had switched to a transport capacity limited or variable erosion-deposition regime during our study period.

The dynamic RUSLE3D simulation carved a deep incision in the main gully channel where water accumulated (Fig. 7). As a detachment capacity limited model RUSLE3D's results were dominated by erosion and thus negative elevation change. It eroded 1480.75 m$^3$ of sediment with no deposition.

The dynamic USPED simulation eroded the banks of the gully and deposited in channels causing the gully grow wider and shallower (Fig. 8). As a transport capacity limited model USPED generated a distributed pattern with both erosion and deposition. Erosion far exceeded deposition with 1235.08 m$^3$ of sediment eroded and 727.46 m$^3$ deposited for a net change of -507.62 m$^3$. While USPED's pattern of elevation change was grainy and fragmented, it captured the process of channel filling and widening expected with a transport capacity limited soil erosion regime.

The steady state SIMWE simulation for a variable erosion-deposition regime predicted the morphological processes and features expected of its regime including gradual aggradation, channel widening, the formation of depositional ridges along the thalweg of the channel, and the development of the depositional fan (Fig. 9). SIMWE was the closest to the observed baseline volumetric change. It balanced erosion and deposition with 785.56 m$^3$ of sediment eroded and 608.91 m$^3$ deposited for a net change of -149.66 m$^3$. Only the SIMWE simulation deposited sediment on the depositional fan. While the difference

of lidar surveys showed that 244.60 $m^3$ of sediment were deposited on the fan, SIMWE predicted that 54.05 $m^3$ would be deposited.

SIMWE was unique in simulating unsteady flows (Fig. 9a) and fine-scale geomorphological processes such as the development of depositional ridges and a depositional fan. While USPED generated a grainy pattern of erosion and deposition, it was much faster than SIMWE (Table 2) and still simulated the key morphological patterns and processes – channel incision, filling, and widening. Given their speed and approximate modeling of erosive processes, RUSLE3D and USPED are effective for simulating landscape evolution on large rasters. RUSLE3D for example has been used to model erosion for the entire 650 $km^2$ Fort Bragg installation at 9 m resolution (Levine et al., 2018).

## 4 Discussion

Limitations of this landscape evolution model include shallow overland flow, units, computation time, and raster size. r.sim.terrain only models shallow overland flows, not fluvial processes or subsurface flows. It requires data – including elevation and rainfall intensity – in metric units. The implementation of SIMWE in GRASS GIS is computationally intensive and may require long computation times even with multithreading. Because SIMWE uses a Green's function Monte Carlo solution of the sediment transport equation, the accuracy, detail, and smoothness of the results depend on the number of random walkers. While a large number of random walkers will reduce the numerical error in the path sampling solution, it will also greatly increase computation time. A customized compilation of GRASS GIS is needed to run SIMWE with more than 7 million random walkers. This limits the size of rasters that can be easily processed with SIMWE, while RUSLE3D and USPED are much faster, computationally efficient, and can easily be run on much larger rasters.

In the future we plan to assess this model by comparing simulations against a monthly timeseries of submeter resolution surveys by unmanned aerial systems and terrestrial lidar. We also plan to develop a case study demonstrating how the model can be used as a planning tool for landscape restoration. Planned enhancements to the model include modeling subsurface flows, accounting for bedrock, and a reverse landscape evolution mode for backward modeling.

## 5 Conclusions

The short-term landscape evolution model r.sim.terrain can simulate the development of gullies, rills, and hillslopes by overland water erosion for a range of hydrologic and soil erosion regimes. The model is novel for simulating landscape evolution based on unsteady flows. The landscape evolution model was tested with a series of simulations for different hydrologic and soil erosion regimes for a highly eroded sub-watershed on Fort Bragg with an active gully. For each regime it generated the morphological processes and features expected. The physics-based SIMWE model simulated morphological processes for a variable erosion-deposition regime such as gradual aggradation, channel widening, scouring, the development of depositional ridges along the thalweg, and the growth of the depositional fan. The empirical RUSLE3D model simulated channel incision in a detachment limited soil erosion regime, while the semi-empirical USPED model simulated channel widening and filling

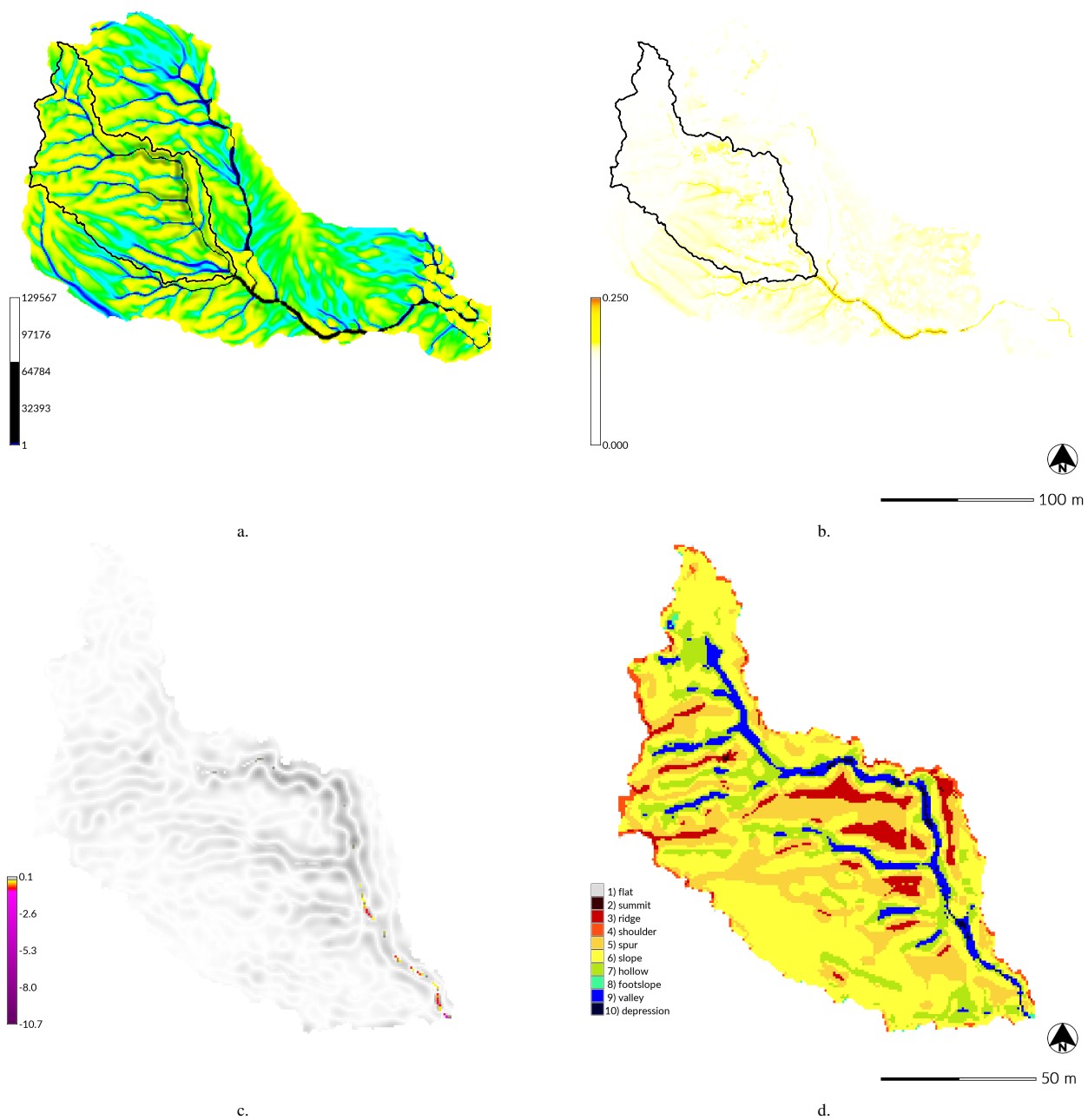

**Figure 7.** Dynamic simulation with RUSLE3D for a 120 min event with a rainfall intensity of 50 mm hr$^{-1}$ for Patterson Branch, Fort Bragg, NC: (a) flow accumulation and (b) erosion [kg m$^{-2}$ s$^{-1}$] for the subwatershed in the final 3 min timestep; (c) net difference [m] and (d) landforms for drainage area 1.

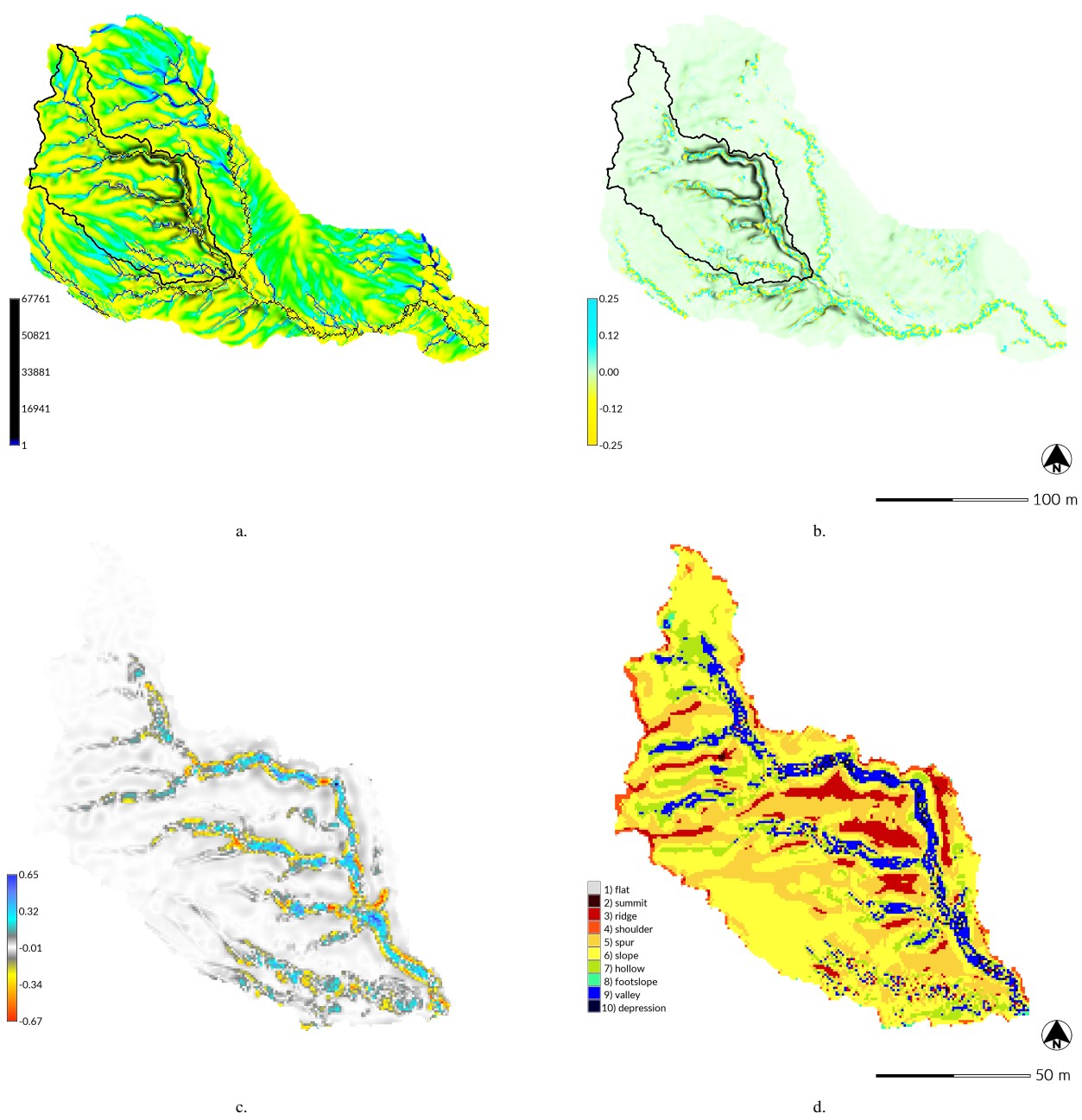

**Figure 8.** Dynamic simulation with USPED for a 120 min event with a rainfall intensity of 50 mm hr$^{-1}$ for Patterson Branch, Fort Bragg, NC: (a) flow accumulation and (b) erosion-deposition [kg m$^{-2}$ s$^{-1}$] for the subwatershed in the final 3 min timestep; (c) net difference [m] and (d) landforms for drainage area 1.

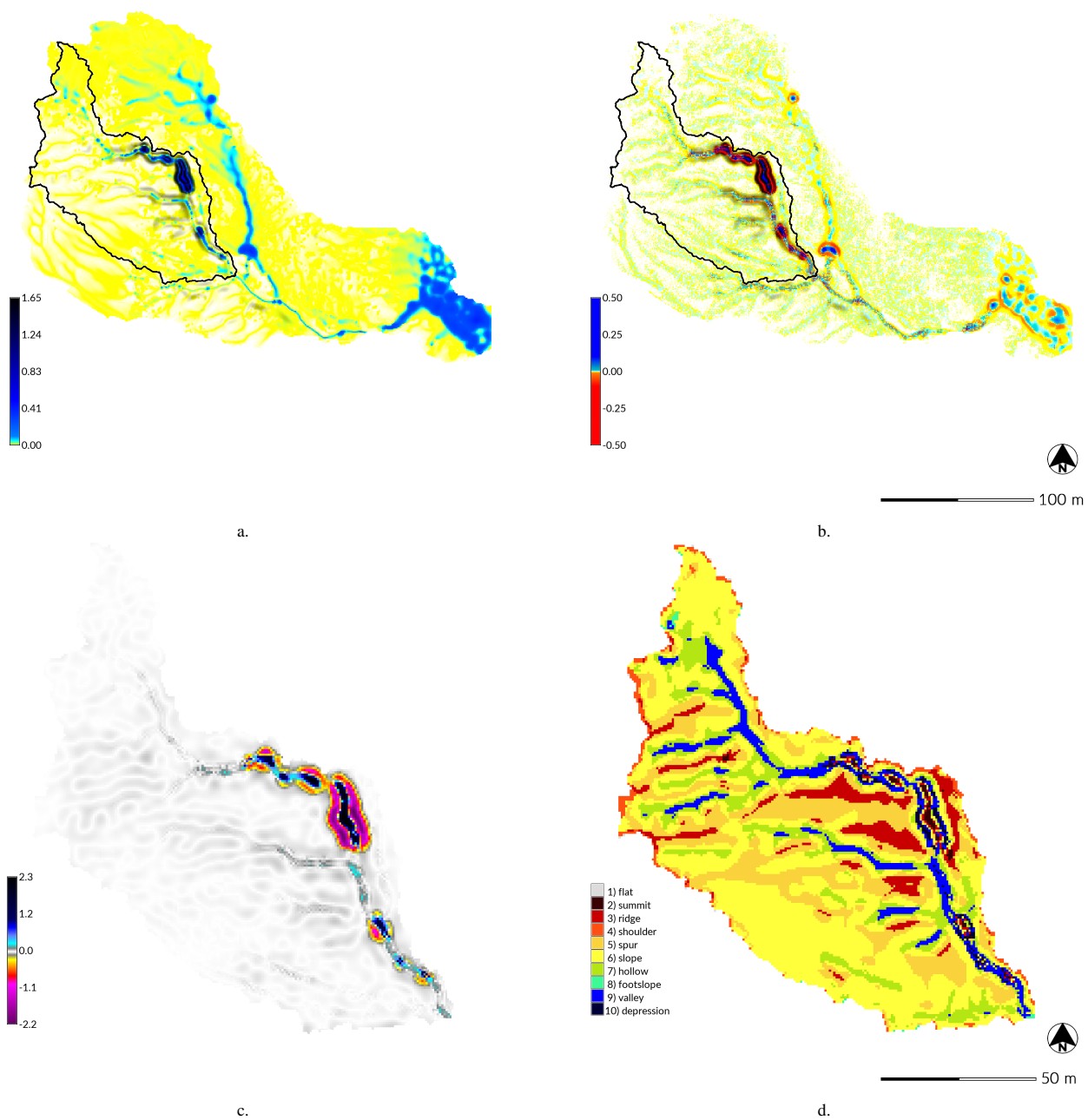

**Figure 9.** Steady state SIMWE simulations for a 120 min event with a rainfall intensity of $50\,\mathrm{mm\,hr^{-1}}$ for Patterson Branch, Fort Bragg, NC: (a) depth of unsteady flow [m] and (b) erosion-deposition $[\mathrm{kg\,m^{-2}\,s^{-1}}]$ for the subwatershed; (c) net difference [m] and (d) landforms for drainage area 1.

in a transport limited regime. Since r.sim.terrain is a GIS-based model that simulates fine-scale morphological processes and features, it can easily and effectively be used in conjunction with other GIS-based tools for geomorphological research, land management and conservation, erosion control, and landscape restoration.

*Code and data availability.* As a work of open science this study is reproducible, repeatable, and recomputable. Since the data, model,

GIS, dependencies are all free and open source, the study can easily be reproduced. The landscape evolution model has been implemented in Python as module for GRASS GIS, a free and open source GIS. The source code for the model is hosted on GitHub at https://github. com/baharmon/landscape_evolution under the GNU General Public License version 2. The code repository also includes Python scripts for running and reproducing the simulations in this paper. The digital object identifier (DOI) for the version of the software documented in this paper is: https://doi.org/10.5281/zenodo.3243699. There are detailed instructions for running this model in the manual at https://grass.osgeo.

org/grass76/manuals/addons/r.sim.terrain.html and the tutorial at https://github.com/baharmon/landscape_evolution/blob/master/tutorial.md. The geospatial dataset for the study area is available on GitHub at https://github.com/baharmon/landscape_evolution_dataset under the Open Database License with the DOI: https://doi.org/10.5281/zenodo.3243700. The data log has a complete record of the commands used to process the sample data. The source code, scripts, data, and results are also hosted on the Open Science Framework at https://osf.io/tf6yb/ with the DOI: https://doi.org/10.17605/osf.io/tf6yb.

*Author contributions.* Brendan Harmon developed the models, code, data, case studies, and manuscript. Helena Mitasova contributed to the development of the models and case studies and revised the manuscript. Anna Petrasova and Vaclav Petras contributed to the development of the code. All authors read and approved the final manuscript.

*Competing interests.* The authors declare that they have no conflict of interest.

*Acknowledgements.* We acknowledge the GRASS GIS Development Community for developing and maintaining GRASS GIS.

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
