# Peer review of "r.sim.terrain 1.0: a landscape evolution model with dynamic hydrology"

_Geoscientific Model Development, 2019_

## Short Comment (SC1) · 14 Mar 2019

Dear authors,

in my role as Executive editor of GMD, I would like to bring to your attention our Editorial version 1.1:

http://www.geosci-model-dev.net/8/3487/2015/gmd-8-3487-2015.html

This highlights some requirements of papers published in GMD, which is also available on the GMD website in the 'Manuscript Types' section:

http://www.geoscientific-model-development.net/submission/manuscript_types.html

In particular, please note that for your paper, the following requirement has not been
met in the Discussions paper:

- "The main paper must give the model name and version number (or other unique identifier) in the title."

Please add a version number for r.sim.terrain to the title upon your revised submission to GMD.

Yours,

Astrid Kerkweg

———————————————————

---

## Referee Comment (RC1) · Anonymous Referee #1 · 20 Mar 2019

Referee comment for:

*r.sim.terrain: a dynamic landscape evolution model*

by B.A. Harmon et al. (2019)

**General comments:**

Overall, the paper presents a novel and interesting model that fills gaps in the current modelling literature around the landscape evolution associated with gully erosion. It outlines the strengths of different models (RUSLE, USPED, and SIMWE) to simulate change at different spatial and temporal scales, and the equations that influence sediment flow and transport, and thus the landscape evolution. A case study is presented showing how the model simulates the development of ephemeral gullies, rills, and hillslopes under the same 120min rainfall event but for different intensities and erosion regimes. This type of model is of interest to the community around Geoscientific Model Development

Although the difference between steady-state and dynamic flow regimes is discussed, the differences between the erosion regimes (e.g. detachment capacity limited, transport capacity limited, erosion-deposition and detachment limited) are less clear. A more thorough discussion of those regimes and their differences would allow for a clearer understanding of the results of the model compared to the typical characteristics associated with these regimes. On P16 L24 to L27, the results of SIMWE were compared to the characteristics typical of the simulated erosion regime. Establishing the characteristics of the erosion regimes earlier, perhaps after the explanation of the flow regimes, would give the reader more clarity regarding what influences these regimes and how the model compares to real-world characteristics.

Given that the study area has information for 2012 and 2016, one possible improvement is to compare the model results to the observed difference between those two years. Although the results section on P16 compares the modelled characteristics with typical erosion regime characteristics, the comparison to the 2012-2016 data is limited to P16 L23. Adding validation of model results against observed landscape evolution would show the strengths of the model.

Another possible improvement is to more clearly present the limitations of the model in their own section. On P4 L22, the model limitation of not modelling fluvial processes is mentioned. By having a clear limitations section with information about model assumptions, the reader is more informed about the model and how it may affect results.

The quality of the figures and the presentation of spatial data is a major issue of the paper. With the exception of Figure 1, many of the figures are too small to be analysed in detail. The legends are pixelated (Figure 4c, 4e, and 4f) or cut off (Figure 5a and 5d). The legends for the landform maps (Figures 5b, 5e, 6b, and 6e) would benefit from the labels presented in Figures 4e and 4f.

The colours chosen for the figures could also be improved. For example, Figure 2b shows a landscape with yellow/orange/blue colours but the colourbar only shows a scale of yellow to orange. Using the hillshade layer seems to darken the colours within the gully and the reader is unable to clearly see those colours.

The use of a 3D top-down view in Figure 5 makes it difficult to see what is occurring within the gully area where the differences are most important. Some figures are presenting differences (Figure 5c, 5f, 6c) that cannot be visualised clearly because most of them are occurring within the gully area and thus "blocked" by the 3D view and hillshade.

Overall, the figures can be improved, especially for visualisation of the key results and differences, and that would contribute to the overall quality of the paper. The differences may be better visualised through 2D top-down view, or 2D cross-sections, or even zooming into the most critical areas of the gully. At the watershed scale and using the current visualisation, the results are difficult to visually interpret and do not supplement the written results well.

**Specific comments:**

- P3, L22: According to Dabney et al. (2014), RUSLER refers to RUSLE2-Raster which is a distributed form of the Revised Universal Soil Loss Equation Version 2, which is normally referred to as RUSLE2. The paper is referring to the Revised Universal Soil Loss Equation Version 2 when it is using the RUSLE2-Raster acronym. Please clarify if the paper is referring to RUSLER or RUSLE2.
- P6, L11 to P7, L6: This paragraph would be better presented in a table or a flowchart showing how the model switches erosion regimes based on rainfall intensity.
- P14, L11 to L13: Additional detail about how the information about K-factor, C-factors, Manning's, and runoff rates were derived would be useful for those who wish to apply the model in their study area.

**Technical corrections:**

- P3, L17: Since LIDAR is an acronym for Light Detection and Ranging, mentions of LIDAR should be in capitals and the first instance should have the accompanying meaning of LIDAR.
- P4, L28 and similar headings: For these headings, referee suggests formatting as follows "Simulation of Water Erosion Model (SIMWE)" and only using the acronym on the following line.
- P6, Table 1: Citation of "(Dennis C. Flanagan et al., 2013)" should just be "(Flanagan et al., 2013)"
- P10, L15: The addition of "($a$)" after "The upslope contributing area per unit width" would allow for a clearer connection to Equation 12.
- P13, L14 and P14, L1: Scientific names should be italicised.

**References:**

Dabney, S., Vieria, D., Bingner, R., Yoder, D., and Altinakar, M.: Modeling Agricultural Sheet, Rill and Ephemeral Gully Erosion, in: ICHE 2014. Proceedings of the 11th International Conference on Hydroscience & Engineering, pp. 1119-1126, Karlsruhe, 2014.

---

## Referee Comment (RC2) · Anonymous Referee #2 · 4 Apr 2019

This paper describes a new landscape evolution model, r.sim.terrain, that runs in the GRASS GIS environment. The authors claim that this model simulates the dynamics of landscape evolution, and particularly gully formation, using any of three different models of overland flow and sediment detachment/transport/deposition. This is a potentially useful simulation model because of its high resolution, potential to simulate erosion/deposition in several different ways, potential to simulate short-term but significant landscape events like gullying, and operation within a widely used, open source GIS environment.

Unfortunately, it is difficult to evaluate the performance of the model for a number of reasons. The text has detailed descriptions of the underlying theory behind many components of the overall simulation model (section 2) but does not indicate how these

relate to one another. While there are many colorful figures of landscape evolution results, there is no representation of the model itself (e.g., UML or other flow-chart-like visualization)–nor is there much in the way of narrative description of how the model actually works to combine the described elements.

Section 3.2 describes several experimental runs with the simulation model. These experiments are summarized in Table 3. However, we have no quantitative information about the results of the experiments. There is no information about whether each experiment was run only once or repeated–or whether repetition is needed or not because each parameter setting produces or does not produce only one outcome. Although the simulation is situated in a realistic setting based on digital data from Ft. Bragg, NC, the rationale for some of the parameterizations are not given, particularly the important rainfall settings. These might be completely reasonable, but the authors should indicate if these are based on empirical rainfall data or have another basis (e.g., extreme values to test the model sensitivity).

The authors have collected detailed, time series, LiDAR and orthophoto data from the test area represented in the simulation. But they make no attempt to compare the simulation experiments with these data in any quantitative sense. Rather they give only brief, subjective assessments of model behavior. It would seem rather easy to compare the model with the empirical data to see which experiments are better or worse fits and in what ways.

Finally, I downloaded and installed r.sim.terrain into GRASS and tried to run it. I strongly commend the authors for making the model code and test data available. This is critically important for research based on modeling like this paper.

I installed r.sim.terrain into the most current version of GRASS according to the directions in the manuscript (i.e., using g.extension). Unfortunately I ran into several problems that made testing the the model impossible. First, there is a link to the test dataset in the model online help, but this link does not work. Using information in the

paper, I was able to go to the GitHub site and poke around until I found the test data set and installed it into my GRASS data directory. I then followed the steps in the tutorial to simply see how it ran–thinking to compare the different overland flow and erosion/deposition methods, and time series vs. event. The command given to test the model failed initially because it was missing the rather critical "elevation" argument. So I added that. Then it started but rapidly bombed with errors related to the time series part. I copy these below. So I never did get it to run.

In sum, I think this can be a valuable simulation environment and contribution to the GRASS GIS geospatial modeling toolset. However, the authors need to do a better job of explaining how the model works and not just the conceptual components included in the model. They also need to provide more information about the four experiments performed and their parameter settings. They need to provide some quantitative evaluation of the model results, including comparison with the empirical data they have collected. Finally, they need to fix some probably minor but annoying bugs in the code available for evaluation. I simply can't give a recommendation to publish a paper about code that does not run.

==== command and errors below =====

r.sim.terrain -f runs=event mode=simwe_mode rain_intensity=50.0 rain_interval=120 rain_duration=10 walkers=1000000 detachment_value=0.01 transport_value=0.0001 manning=mannings runoff=runoff elevation=fortbragg_elevation_10m_2012@PERMANENT

100%

100%

100%

100%

100%

100%

100%

WARNING: Overwriting space time raster dataset <elevation_timeseries> and unregistering all maps

WARNING: Overwriting space time raster dataset <depth_timeseries> and unregistering all maps

WARNING: Overwriting space time raster dataset <erdep_timeseries> and unregistering all maps

WARNING: Overwriting space time raster dataset <flux_timeseries> and unregistering all maps

WARNING: Overwriting space time raster dataset <difference_timeseries> and unregistering all maps

Gathering map information...

ERROR: Unable to insert dataset <fortbragg_elevation_10m_2012@PERMANENT> of

type raster in the temporal database. The mapset of the dataset does not match the current mapset

Traceback (most recent call last):

File "/Users/cmbarton/Library/GRASS/7.7/Modules/scripts/r.sim.terrain", line 2698, in <module> main()

File "/Users/cmbarton/Library/GRASS/7.7/Modules/scripts/r.sim.terrain", line 607, in main elevation = dynamics.rainfall_event()

File "/Users/cmbarton/Library/GRASS/7.7/Modules/scripts/r.sim.terrain", line 1903, in rainfall_event overwrite=True)

File "/Applications/GRASS-7.7.app/Contents/Resources/etc/python/grass/script/core.py",
line 440, in run_command return handle_errors(returncode, returncode, args, kwargs)

File "/Applications/GRASS-7.7.app/Contents/Resources/etc/python/grass/script/core.py",
line 342, in handle_errors returncode=returncode)

grass.exceptions.CalledModuleError:     Module     run     None     t.register     –o     -
i     maps=fortbragg_elevation_10m_2012@PERMANENT     type=raster     in-
put=elevation_timeseries increment=120 minutes start=2000-01-01 00:00:0

---

## Referee Comment (RC3) · Anonymous Referee #3 · 8 Apr 2019

This manuscript presents a useful new erosion modeling package. Some of the nice aspects of this contribution include:

- The package brings together a collection of alternative erosion and transport laws, which allows for inter-model comparison: an especially valuable thing given that the community does not seem to have a consensus on what are the 'right' rules to use in any particular setting. Furthermore, the manuscript provides a nice proof-of-concept demonstration in using this package to compare the predictions of different process formulations.

- As the title suggests, a novel feature of this modeling package is the capability of representing dynamic hydrology: that is, a representation of time-varying overland flow. As the authors note, this capability is not present in most landform evolution models

(though there are a few that have tried to honor it in some form or another).

- The code and data are open source, and maintained in a version-controlled online repository. In fact, the authors have gone even further than this commendable practice by making their examples fully reproducible. Bravo!

- The package is embedded in the open-source GRASS GIS, which makes it easier to provide geospatial input data, and to analyze and display model output.

In my view, this is a very nice contribution to the soil-modeling and landscape-modeling ecosystem, and in terms of reproducibility, it sets the bar high for future modelers.

The main area for improvement of the manuscript, in my view, lies in the presentation of the governing equations. In several places that I have noted below, the relationship between different equations is murky, and there are places where the units seem to be inconsistent. I think these issues should be straightforward to address.

Detailed comments, keyed to page and line number (or in some places, equation, table, or figure number):

1/title - This is a total quibble, and please feel free to ignore it, but my first reaction to 'dynamic landscape evolution model' was to ask (rhetorically) 'is there any other kind'? Consider 'landscape evolution model with dynamic hydrology' as an alternative (admittedly a less pithy one).

1/5 'steady state or dynamic model' could be read as implying that the entire model is steady state, not just the surface water flow rates. Suggest re-wording: 'using either a steady state or dynamic representation of overland flow, ...'

2/2 I agree with the sentiment, but suggest rewording to 'a landscape evolution model that includes time-evolving surface water discharge', to avoid confusion over which aspect of the model is dynamic.

3/8 The phrase 'until water flow reaches steady state' suggests that the positive feedback (presumably between deepening/widening and attraction of more surface water flow) stops at this point. I don't think that is necessarily true; you could have a feedback between morphology and flow under steady runoff too.

3/11 Please explain what is meant by erosion-deposition regime.

3/14 Detachment vs transport capacity: this sounds backwards...

3/18-19 There are plenty of other papers that could be cited here, in which one or more of the listed methods was used to study gully erosion. (For example, here's a review paper that cites some TLS applications to gully erosion: Telling, J., Lyda, A., Hartzell, P., & Glennie, C. (2017). Review of Earth science research using terrestrial laser scanning. Earth-Science Reviews, 169, 35-68.)

Figure 2: please give location in caption. Also, numbers on color bars and scale bar are barely legible.

6/2 typo

6/2 I guess 'partial derivatives of the topography' means a numerical approximation of the derivative of the elevation field with respect to the two cardinal grid directions. Recommend more precision in wording here.

6/9 'steady state dynamics' - I think I understand what you mean here, but the phrase itself is awkward (it is self-contradictory)

Table 2: this is only a partial list of codes that have been published in, say, the last ten years. Why choose these particular ones?

Be careful about giving the spatial scale for these models. At least some of these codes have been used and published at a variety of different spatial scales, from say the size of a rilled hillslope to that of a small country; and in some cases (e.g., SIBERIA) is sometimes presented in a dimensionless mode in which no spatial scale at all is given or implied. As to temporal scale, I thought that at least some of these can also be run

in 'event' mode.

Also, my understanding is that Landlab is not itself a model, but rather is a programming library that contains components that can be used to build various types of model, including landscape evolution. That said, people seem to have built landscape evolution models using Landlab (the Landlab website lists some of these). Maybe it would make sense to label this entry as 'Landlab-built erosion models' or something like that.

Section 2.1 generally: I like the way that this is carefully organised into sub-sections. However, the order of presentation confused me. Often, authors presenting a set of governing equations will start with the high-level conservation law(s), and then define each term more precisely. As noted below, there's an opportunity to do this at least partly in subsection 2.1.1.

Equation 2: it would be helpful to give some context and referencing. I think this idea comes from Foster and Meyer (1972), right? If I remember correctly, their key assumption was that the ratio of transport rate to transport capacity, plus the ratio of detachment rate to detachment capacity, sum to unity. Assuming I did the math right, this leads to a first-order reaction-like equation:

dz/dt = ds = sigma (qs - Tc)

I recommend presenting it this way here in section 2.1.1 (in addition to the definition given in eq 2), because this relates transport and detachment to the rate of change of elevation, and motivates the need for definitions for qs, Tc, and Dc.

Note that there seems to be a problem with units in one of the factors in eq 2: if Tc and Dc had the same units (as is listed), then sigma would be dimensionless. I suspect Dc is actually in kg m^-2 s^-1 (detached mass per unit area per time).

Equation 4: symbol v is used without being introduced. Presumably it is the depth-averaged flow velocity vector in (x,y). Either define v or use q (which you've defined already).

Also, whereas the paper is premised on the value of having a dynamic representation of surface-water hydrology (which eq 3 is), equation 4 is actually a steady solution, is it not? If the model indeed uses a fully time-varying flow model, the equations presented in this sub-section should show this. In addition, it would be helpful to provide a reference for this form of the diffusion-wave approximation (could be to a hydrology text that gives the derivation and assumptions).

Please give units of epsilon.

8/10 suggest specifying '...density in the water column', so it is clear that this is a mass concentration rather than a bulk density of resting sediment.

8/15 'steady state sediment flow with diffusion' - I'm confused by this. The equation is time-dependent, so how is it steady state? And the definition of qs above is advective, not diffusive.

8/17 So we need a definition for ds, which as suggested above, you could provide in section 2.1.1.

8/23 In the previous equation, you used a continuum formulation, whereas here you're giving a discretized-in-time form. Please be consistent. I suggest sticking with continuum forms, because these don't require you to make any statements about numerical approximation. And in fact, as noted above, I recommend putting equation 7 in section 2.1.1.

Equation 8: this equation is not dimensionally consistent. If you write it in continuum form,

dz/dt = -(1/rho) qs

you have m/s on the left and m2/s on the right. I'm also not convinced that the equation expresses the idea you want. I'm guessing that a detachment-limited regime would look more like dz/dt = -(1/rho) Dc. Then it becomes a question of what is your detachment capacity law? You've already introduced detachment capacity in Dc = sigma

[Figure]

Tc (eq 2). In order to close the equations, you need either a definition of Dc or Tc. Presumably these depend in some fashion on water discharge or velocity or boundary shear stress. Please specify (or, if I have misunderstood, explain why the equation set given is sufficient to describe the SIMWE module). Actually, after reading farther in the manuscript, I think the idea is that the RUSLE equation can be used for Dc in detachment-limited mode. If that's correct, then say something to the effect that the definition of Dc will be given in section so-and-so, and then use the symbol Dc in that section.

Regarding the role of qs, I suspect what you're after is the notion that qs is the upstream/upslope integral of ds, is that right? If so, it would be helpful to present the math.

9/21 please give the functional form of this relationship

10/2 I get that there's a long tradition of practical empiricism in soil-erosion research. But what about pushing ever so gently back on it by presenting equation 10 in a slightly less brutally ugly form? Something like:

er / e_ref = 1 - a exp( -ir / i_ref )

where e_ref is reference energy equal to ... and i_ref is reference rainfall intensity equal to ...

10/7 shouldn't this be rainfall depth rather than volume?

Equation 11: again the units seem to be off here (apart from the oddity of having an 'index' that has [weird] units). I get the right side as being:

MJ haˆ-1 mmˆ-1 x mm x s = MJ haˆ-1 s???

11/3 the subsection is called 'Sediment flow' but it reads like an erosion rate. Though I guess it works given that you're defining it as mass flow per time per area.

Equation 13: again I'm struggling with units. I get:

(MJ mm haˆ-1 hrˆ-1) x (ton ha hr haˆ-1 MJˆ-1 mmˆ-1) = (haˆ-1) x (ton)

which are not the units given for E.

11/16-17 it's not clear to me how these equations relate. Maybe you mean that the definition of E in equation 13 is the SAME AS ds (or -ds) for transport-limited conditions, and Dc for detachment-limited conditions? In that case, it might suffice to simply call equation 13 the definition of Dc. You could then give the defintion of Tc as

sigma = Dc/Tc ==> Tc = Dc/sigma

(Note: it would be more intuitive to think in terms of a length scale, L = 1 / sigma, which is then the characteristic distance over which steady, uniform overland flow reaches its carrying capacity on a planar slope).

11/29 not clear to me what 'topographic component of overland flow' means

Equation 15: is T the same as Tc? Also, again, I'm not sure the units are correct here, please check, and correct if necessary.

16/1 intriguing comment about positive feedback loops, can you explain more?

Figures 5 and 6: why the different color schemes in two of the three comparisons (top and bottom rows)?

Figure 6: if the figure is meant to compare runs with two different rainfall intensities, which intensity was used for the upper and middle figures?

Software: I tested the model software by installing the latest stable release of GRASS GIS, then going to the GitHub repository for the model's source code. By following the "Basic Instructions" listed there, I was able to install the r.sim.terrain extension and run the example.

---

## Author Comment (AC1) · 18 Apr 2019

Could you please re-run the model if you have time? We have updated the documentation both on the manual page and the GitHub repository with the tutorial. Your run of the model failed due to incomplete documentation rather than bugs. The input elevation raster must be in the current mapset (for registration in the temporal database), so it should be copied from the PERMANENT mapset to the current working mapset before the model is run.

We have added a section with basic instructions to the manual page (https://grass.osgeo.org/grass76/manuals/addons/r.sim.terrain.html) and to the repository readme (https://github.com/baharmon/landscape_evolution). We have also written

a longer tutorial (https://github.com/baharmon/landscape_evolution/blob/master/tutorial.md) that details running each model with instructions and examples for RUSLE3D, USPED, and detachment limited, transport limited, and variable erosion-deposition regimes with SIMWE.

A new concept diagram for r.sim.terrain is attached. Please let us know if you have any comments or critiques.

―――――――――――――――――――

**r.sim.terrain**

input data

determine
model

RUSLE3D     SIMWE     USPED

*iterate*                                    *iterate*

determine
regime

detachment     variable        transport
limited        erosion-        limited
               deposition

*iterate*

output
data

**Fig. 1.** Concept diagram for r.sim.terrain

---

## Author Comment (AC2) · 15 May 2019

Since we have restructured the manuscript, the references to equations and sections below refer to the discussion paper. The revised paper will have different sections, equations numbers, figures, and tables due to our revisions.

[Figure]

**1 Reviewer 1**

1.1

Although the difference between steady-state and dynamic flow regimes is discussed, the differences between the erosion regimes (e.g. detachment capacity limited, transport capacity limited, erosion-deposition and detachment limited) are less clear. A more thorough discussion of those regimes and their differences would allow for a clearer understanding of the results of the model compared to the typical characteristics associated with these regimes. On P16 L24 to L27, the results of SIMWE were compared to the characteristics typical of the simulated erosion regime. Establishing the characteristics of the erosion regimes earlier, perhaps after the explanation of the flow regimes, would give the reader more clarity regarding what influences these regimes and how the model compares to real-world characteristics.

We have restructured the paper and now thoroughly discuss soil erosion-deposition regimes in Section 2.1.2 including equations 7-11.

1.2

Given that the study area has information for 2012 and 2016, one possible improvement is to compare the model results to the observed difference between those two years. Although the results section on P16 compares the modelled characteristics with typical erosion regime characteristics, the comparison to the 2012-2016 data is limited to P16 L23. Adding validation of model results against observed landscape evolution would show the strengths of the model.

This is a model description paper, rather than a model evaluation paper. While we have added a quantitative comparison with linear regressions and bivariate scatterplots of topographic change to the paper, we plan to conduct a rigorous quantitative evaluation of r.sim.terrain in future work. Because the models are for different erosion regimes, different study sites each with a different dominant regime would be needed to quantitatively assess each model against a relevant baseline. A more accurate, higher frequency of high resolution topographic surveying is also needed, ie. monthly surveys with terrestrial lidar or unmanned aerial systems. We have added plans for future work to the new Discussion section.

1.3

Another possible improvement is to more clearly present the limitations of the model in their own section. On P4 L22, the model limitation of not modelling fluvial processes is mentioned. By having a clear limitations section with information about model assumptions, the reader is more informed about the model and how it may affect results

We have added a paragraph on the limitations of the model to a new Discussion section.

1.4

The quality of the figures and the presentation of spatial data is a major issue of the paper. With the exception of Figure 1, many of the figures are too small to be analysed in detail. The legends are pixelated (Figure 4c, 4e, and 4f) or cut off (Figure 5a and 5d). The legends for the landform maps (Figures 5b, 5e, 6b, and 6e) would benefit from the labels presented in Figures 4e and 4f. The colours chosen for the figures could also be improved. For example, Figure 2b shows a landscape with yellow/orange/blue colours but the colour bar only shows a scale of yellow to orange. Using the hillshade layer seems to darken the colours within the gully and the reader is unable to clearly see those colours. The use of a 3D top-down view in Figure 5 makes it difficult to see

what is occurring within the gully area where the differences are most important. Some figures are presenting differences (Figure 5c, 5f, 6c) that cannot be visualised clearly because most of them are occurring within the gully area and thus "blocked" by the 3D view and hillshade. Overall, the figures can be improved, especially for visualisation of the key results and differences, and that would contribute to the overall quality of the paper. The differences may be better visualised through 2D top-down view, or 2D cross-sections, or even zooming into the most critical areas of the gully. At the watershed scale and using the current visualisation, the results are difficult to visually interpret and do not supplement the written results well.

All figures have been redone. They have newer higher resolution legends, scale bars, and north arrows. Figures 3, 5, 6, and 7 are zoomed in on a drainage area (Drainage Area 1) within the subwatershed. These figures are now focused on the main channel of the gully and should show more legible detail. Figure 4b shows the drainage areas. In addition to zooming in on Drainage Area 1, Figures 5, 6, and 7 are now presented in 2x2 columns and rows, rather than 2x3 columns and rows so that the images are larger and more detail is visible. Selected figures are now 2D rather than 3D maps. We have changed or removed hill shading from select maps to improve their legibility. We have also added bivariate scatterplots of elevation change.

1.5

P3, L22: According to Dabney et al. (2014), RUSLER refers to RUSLE2-Raster which is a distributed form of the Revised Universal Soil Loss Equation Version 2, which is normally referred to as RUSLE2. The paper is referring to the Revised Universal Soil Loss Equation Version 2 when it is using the RUSLE2-Raster acronym. Please clarify if the paper is referring to RUSLER or RUSLE2.

We have replaced this with: "Gully erosion has been simulated with RUSLE2-Raster (RUSLER) in conjunction with the Ephemeral Gully Erosion Estimator (EphGEE) (Dabney et al., 2014)"

1.6

P6, L11 to P7, L6: This paragraph would be better presented in a table or a flowchart
showing how the model switches erosion regimes based on rainfall intensity

To more clearly present how soil erosion regimes are handled in this model we have
added Section 2.1.2 Erosion-Deposition Regimes. We have also removed the detach-
ment limited and transport limited cases for SIMWE to avoid unnecessary complexity
in the paper and results.

1.7

P14, L11 to L13: Additional detail about how the information about K-factor, C-factors,
Manning's, and runoff rates were derived would be useful for those who wish to apply
the model in their study area.

We have added links to detailed instructions for deriving these maps in the tutorial. We
have also added a link to our data log with a complete record of the commands used
to process the sample data.

1.8

P3, L17: Since LIDAR is an acronym for Light Detection and Ranging, mentions of LI-
DAR should be in capitals and the first instance should have the accompanying mean-
ing of LIDAR.

We have followed the recommendation in the paper "Let's agree on the casing of lidar" (
Deering  Stoker, 2014), which shows that 65% of literature uses lidar (including USGS),

while 17% use LIDAR and 14% use LiDAR. Their reasoning is that "lidar" is the most common usage, the original usage, and the usage recommended by style manuals However, the latest issues of ISPRS journals use LiDAR, so we will defer to the journal editor on what their standard should be. For our part we prefer to follow USGS usage of lidar.

Deering, Carol  Stoker, Jason. (2014). Let's agree on the casing of lidar. Lidar Magazine. 4. 48-51. http://lidarmag.com/wp-content/uploads/PDF/LiDARNewsMagazine_DeeringStoker-CasingOfLiDAR_Vol4No6.pdf

1.9

P4, L28 and similar headings: For these headings, referee suggests formatting as follows "Simulation of Water Erosion Model (SIMWE)" and only using the acronym on the following line.

Reformatted as recommended.

1.10

P6, Table 1: Citation of "(Dennis C. Flanagan et al., 2013)" should just be "(Flanagan et al., 2013)"

Citation fixed as recommended.

1.11

P10, L15: The addition of "(a)" after "The upslope contributing area per unit width" would allow for a clearer connection to Equation 12.

Variable added as recommended.

1.12

P13, L14 and P14, L1: Scientific names should be italicised.

Scientific names have been italicized.

**2 Reviewer 2**

2.1

The text has detailed descriptions of the underlying theory behind many components of the overall simulation model (section 2) but does not indicate how these relate to one another. While there are many colorful figures of landscape evolution results, there is no representation of the model itself (e.g., UML or other flow-chart-like visualization) - nor is there much in the way of narrative description of how the model actually works to combine the described elements

We have added a conceptual diagram of the model as Figure 2.

2.2

Section 3.2 describes several experimental runs with the simulation model. These experiments are summarized in Table 3. However, we have no quantitative information about the results of the experiments. There is no information about whether each experiment was run only once or repeated–or whether repetition is needed or not because each parameter setting produces or does not produce only one outcome. Al-

**GMDD**

though the simulation is situated in a realistic setting based on digital data from Ft. Bragg, NC, the rationale for some of the parameterizations are not given, particularly the important rainfall settings. These might be completely reasonable, but the authors should indicate if these are based on empirical rainfall data or have another basis (e.g., extreme values to test the model sensitivity).

Repeat runs are not needed because RUSLE and USPED are empirical, non-stochastic models and produce only one outcome. SIMWE path sampling method includes stochastic component for solving the continuity equations but this relates to the accuracy of the solution, e.g., a high number of walkers reduces the numerical error associated with the path sampling solution. We have added a discussion of this to the paragraph on limitations. We have added more information about the parameters of the simulations in subsections 3.1 and 3.2 on the study site and simulations. This includes a link to detailed instructions for the sample data and a discussion of the design storms and their rationale.

2.3

The authors have collected detailed, time series, LiDAR and orthophoto data from the test area represented in the simulation. But they make no attempt to compare the simulation experiments with these data in any quantitative sense. Rather they give only brief, subjective assessments of model behavior. It would seem rather easy to compare the model with the empirical data to see which experiments are better or worse fits and in what ways.

We consider this manuscript a model description paper, rather than a model evaluation paper. While we have added a quantitative comparison with linear regressions and bivariate scatterplots of topographic change to the paper, we plan to conduct a rigorous quantitative evaluation of r.sim.terrain in future work. Because the models are for different erosion regimes, different study sites with a different dominant regimes may

be needed to quantitatively assess each model against a relevant baseline. A higher frequency of high resolution topographic surveying would also help, ie. monthly surveys with terrestrial lidar or unmanned aerial systems. We have added plans for future work to the conclusion.

**2.4**

Finally, I downloaded and installed r.sim.terrain into GRASS and tried to run it. I strongly commend the authors for making the model code and test data available. This is critically important for research based on modeling like this paper. I installed r.sim.terrain into the most current version of GRASS according to the directions in the manuscript (i.e., using g.extension). Unfortunately I ran into several problems that made testing the the model impossible. First, there is a link to the test dataset in the model online help, but this link does not work. Using information in the paper, I was able to go to the GitHub site and poke around until I found the test data set and installed it into my GRASS data directory. I then followed the steps in the tutorial to simply see how it ran–thinking to compare the different overland flow and erosion/deposition methods, and time series vs. event. The command given to test the model failed initially because it was missing the rather critical "elevation" argument. So I added that. Then it started but rapidly bombed with errors related to the time series part. I copy these below. So I never did get it to run.

We have updated the documentation both on the manual page and the GitHub repository with the tutorial. The reviewer's run of the model failed due to incomplete documentation rather than bugs. The input elevation raster must be in the current mapset (for registration in the temporal database), so it should be copied from the PERMANENT mapset to the current working mapset before the model is run. We have added a section with basic instructions to the manual page (https://grass.osgeo.org/grass76/manuals/addons/r.sim.terrain.html) and to the repository readme

(https://github.com/baharmon/landscape_evolution). We have also written a longer tutorial (https://github.com/baharmon/landscape_evolution/blob/master/tutorial.md) that details running each model with instructions and examples for RUSLE3D, USPED, and SIMWE.

2.5

However, the authors need to do a better job of explaining how the model works and not just the conceptual components included in the model. They also need to provide more information about the four experiments performed and their parameter settings. They need to provide some quantitative evaluation of the model results, including comparison with the empirical data they have collected. Finally, they need to fix some probably minor but annoying bugs in the code available for evaluation.

We have added a conceptual diagram of the model and rewritten and expanded the subsection on the simulations and their parameters. We have added quantitative evaluation and comparison of the models and explained its limitations. We, however, would like to reiterate that this is a model description rather than evaluation paper. Finally, while there were no bugs in the code, we have fixed and expanded the documentation, which was incomplete.

**3 Reviewer 3**

3.1

1/title - This is a total quibble, and please feel free to ignore it, but my first reaction to 'dynamic landscape evolution model' was to ask (rhetorically) 'is there any other kind'? Consider 'landscape evolution model with dynamic hydrology' as an alternative

(admittedly a less pithy one).

Titled changed to "r.sim.terrain: a landscape evolution model with dynamic hydrology" as recommended.

3.2

1/5 'steady state or dynamic model' could be read as implying that the entire model is steady state, not just the surface water flow rates. Suggest re-wording: 'using either a steady state or dynamic representation of overland flow ...'

Reworded as recommended.

3.3

2/2 I agree with the sentiment, but suggest rewording to 'a landscape evolution model that includes time-evolving surface water discharge', to avoid confusion over which aspect of the model is dynamic.

Replaced with "A landscape evolution model with dynamic water and sediment flow..."

3.4

3/8 The phrase 'until water flow reaches steady state' suggests that the positive feedback (presumably between deepening/widening and attraction of more surface water flow) stops at this point. I don't think that is necessarily true; you could have a feedback between morphology and flow under steady runoff too

An excellent point. Revised simply by cutting "until water flow reaches steady state."

3.5

3/11 Please explain what is meant by erosion-deposition regime.

We have added a new Section 2.1.2 describing erosion-deposition regimes. This includes equations 7-11.

3.6

3/14 Detachment vs transport capacity: this sounds backwards. . .

Detachment capacity and transport capacity are now more clearly explained in Section 2.1.2 Erosion-Deposition Regimes.

3.7

3/18-19 There are plenty of other papers that could be cited here, in which one or more of the listed methods was used to study gully erosion. (For example, here's a review paper that cites some TLS applications to gully erosion: Telling, J., Lyda, A., Hartzell, P., Glennie, C. (2017). Review of Earth science research using terrestrial laser scanning. Earth-Science Reviews, 169, 35-68.)

This paragraph was just meant to be a brief overview of methods, not a comprehensive review, but more sources is better. We will cite more papers - especially review papers - here.

[Figure]

3.8

Figure 2: please give location in caption. Also, numbers on color bars and scale bar are barely legible.

We added the location to the caption and redone the legends and scale bars at higher resolution.

3.9

6/2 typo

The typo has been fixed.

3.10

6/2 I guess 'partial derivatives of the topography' means a numerical approximation of the derivative of the elevation field with respect to the two cardinal grid directions. Recommend more precision in wording here.

We have more clearly phrased this, explained briefly how it is computed, and added a reference to the chapter Geomorphometry in GRASS GIS (Hofierka et al., 2009) that explains the math and implementation.

3.11

6/9 'steady state dynamics' - I think I understand what you mean here, but the phrase itself is awkward (it is self-contradictory)

Replaced with: "The model simulates dynamic flow regimes when the time step is less

than the travel time for a drop of water or a particle of sediment to cross the landscape. With longer time steps the model simulates a steady state regime."

3.12

Table 2: this is only a partial list of codes that have been published in, say, the last ten years. Why choose these particular ones?

We removed the table and instead list or briefly discuss these landscape evolution models in the text.

3.13

Table 2: Be careful about giving the spatial scale for these models. At least some of these codes have been used and published at a variety of different spatial scales, from say the size of a rilled hillslope to that of a small country; and in some cases (e.g., SIBERIA) is sometimes presented in a dimensionless mode in which no spatial scale at all is given or implied. As to temporal scale, I thought that at least some of these can also be run in 'event' mode. Also, my understanding is that Landlab is not itself a model, but rather is a programming library that contains components that can be used to build various types of model, including landscape evolution. That said, people seem to have built landscape evolution models using Landlab (the Landlab website lists some of these). Maybe it would make sense to label this entry as 'Landlab-built erosion models' or something like that.

We removed the table and instead list or briefly discuss these libraries and landscape evolution models in the text.

3.14

Section 2.1 generally: I like the way that this is carefully organised into sub-sections. However, the order of presentation confused me. Often, authors presenting a set of governing equations will start with the high-level conservation law(s), and then define each term more precisely. As noted below, there's an opportunity to do this at least partly in subsection 2.1.1.

We agree and we will start this description with the general equation for change in elevation (continuous form of eq. 7) followed by general equation for $d_s$. See e.g. eq. (9) in Mitasova et al. 2005 (D(r,t) is our $d_s$).

3.15

Equation 2: it would be helpful to give some context and referencing. I think this idea comes from Foster and Meyer (1972), right? If I remember correctly, their key assumption was that the ratio of transport rate to transport capacity, plus the ratio of detachment rate to detachment capacity, sum to unity. Assuming I did the math right, this leads to a first-order reaction-like equation: dz/dt = ds = sigma (qs - Tc) I recommend presenting it this way here in section 2.1.1 (in addition to the definition given in eq 2), because this relates transport and detachment to the rate of change of elevation, and motivates the need for definitions for qs, Tc, and Dc. Note that there seems to be a problem with units in one of the factors in eq 2: if Tc and Dc had the same units (as is listed), then sigma would be dimensionless. I suspect Dc is actually in kg mĘ̈-2 sĘ̈-1 (detached mass per unit area per time).

We will add a more complete explanation of the Foster and Meyer relationship and the related parameters and fix the units for $D_c$ (eq. 12-13 in Mitasova et al. 2005).

3.16

Equation 4: symbol v is used without being introduced. Presumably it is the depth averaged flow velocity vector in (x,y). Either define v or use q (which you've defined already).

We modified the equation according to the reviewer's suggestion.

3.17

Also, whereas the paper is premised on the value of having a dynamic representation of surface-water hydrology (which eq 3 is), equation 4 is actually a steady solution, is it not? If the model indeed uses a fully time-varying flow model, the equations presented in this sub-section should show this. In addition, it would be helpful to provide a reference for this form of the diffusion-wave approximation (could be to a hydrology text that gives the derivation and assumptions).

The in-depth explanation of mathematical foundations for the shallow water flow simulation has been addressed in several previously published papers, for example Mitasova et al. 2005, and we tried to avoid repeating text presented there, unfortunately this makes some of the concepts and reasoning behind the methods less clear. Therefore, we have rewritten the entire section 2 including more detailed explanation of the method and more specific references.

3.18

Please give units of epsilon.

Units for epsilon are now given in the text.

3.19

8/10 suggest specifying '...density in the water column', so it is clear that this is a mass concentration rather than a bulk density of resting sediment.

Changed as recommended by reviewer.

3.20

8/15 "steady state sediment flow with diffusion' - I'm confused by this. The equation is time-dependent, so how is it steady state? And the definition of qs above is advective, not diffusive.

We will add the equation with diffusion term used in the path sampling solution, see e.g. Mitasova et al. 2005 eq. 16.

3.21

8/17 So we need a definition for ds, which as suggested above, you could provide in section 2.1.1.

Agreed. We will rewrite section 2.1 and include a definition of $d_s$.

3.22

8/23 In the previous equation, you used a continuum formulation, whereas here you're giving a discretized-in-time form. Please be consistent. I suggest sticking with continuum forms, because these don't require you to make any statements about numerical approximation. And in fact, as noted above, I recommend putting equation 7 in section

2.1.1.

We have rewritten the equation into continuum form and moved the entire 2.1.5 section as section 2.1.1 following the reviewers recommendation

3.23

Equation 8: this equation is not dimensionally consistent. If you write it in continuum form, dz/dt = -(1/rho) qs you have m/s on the left and m2/s on the right. I'm also not convinced that the equation expresses the idea you want. I'm guessing that a detachment-limited regime would look more like dz/dt = -(1/rho) Dc. Then it becomes a question of what is your detachment capacity law? You've already introduced detachment capacity in Dc = sigma C5 GMDD Interactive comment Printer-friendly version Discussion paper Tc (eq 2). In order to close the equations, you need either a definition of Dc or Tc. Presumably these depend in some fashion on water discharge or velocity or boundary shear stress. Please specify (or, if I have misunderstood, explain why the equation set given is sufficient to describe the SIMWE module). Actually, after reading farther in the manuscript, I think the idea is that the RUSLE equation can be used for Dc in detachment-limited mode. If that's correct, then say something to the effect that the definition of Dc will be given in section so-and-so, and then use the symbol Dc in that section. Regarding the role of qs, I suspect what you're after is the notion that qs is the upstream/upslope integral of ds, is that right? If so, it would be helpful to present the math.

You are correct, using $q_s$ here was an oversight - eq. 8 is not really needed, because $d_s$ in DLC is erosion rate given by eq 13 which is detachment rate (soil loss) not sediment flow. We have removed this equation.

3.24

9/21 please give the functional form of this relationship

The equation for this relationship is given in the section 2.2.3

3.25

10/2 I get that there's a long tradition of practical empiricism in soil-erosion research. But what about pushing ever so gently back on it by presenting equation 10 in a slightly less brutally ugly form? Something like: er /

Thank you for the suggestion. We will modify the equation accordingly.

3.26

10/7 shouldn't this be rainfall depth rather than volume? Equation 11: again the units seem to be off here (apart from the oddity of having an 'index' that has [weird] units). I get the right side as being: MJ ha-1 mm-1 x mm x s = MJ ha-1 s?

We have revised this equation, checked the units, and introduced it with Equation 2 from Panagos et al. 2015 from which it is derived.

3.27

11/3 the subsection is called 'Sediment flow' but it reads like an erosion rate. Though I guess it works given that you're defining it as mass flow per time per area

You are right - it is an erosion rate (soil loss: mass per area per time). In this paper we were using the term sediment flow for sediment flow per unit width mass per length

per time). We have retitled this subsection as Detachment Limited Erosion Rate and replaced sediment flow with erosion rate throughout the paper for this case.

3.28

Equation 13: again I'm struggling with units. I get: (MJ mm ha-1 hr-1) x (ton ha hr ha-1 MJ-1 mm-1) = (ha-1) x (ton) which are not the units given for E.

We have revised the units for E and R.

3.29

11/16-17 it's not clear to me how these equations relate. Maybe you mean that the definition of E in equation 13 is the SAME AS ds (or -ds) for transport-limited conditions, and Dc for detachment-limited conditions? In that case, it might suffice to simply call equation 13 the definition of Dc. You could then give the definition of Tc as sigma = Dc/Tc ==> Tc = Dc/sigma (Note: it would be more intuitive to think in terms of a length scale, L = 1 / sigma, which is then the characteristic distance over which steady, uniform overland flow reaches its carrying capacity on a planar slope).

We have renamed this section and revised the text following the reviewer suggestion. The equation 8 was removed (see the related answer above)

3.30

11/29 not clear to me what 'topographic component of overland flow' means

It should be sediment flow. We will reword the sentence to clarify.

[Figure]

3.31

Equation 15: is $T$ the same as $T_c$? Also, again, I'm not sure the units are correct here, please check, and correct if necessary.

Yes, $T$ is $T_c$. We will unify the symbols, $R_e$ is missing per year and will fix the units .

3.32

16/1 intriguing comment about positive feedback loops, can you explain more?

We will add an explanation.

3.33

Figures 5 and 6: why the different color schemes in two of the three comparisons (top and bottom rows)?

We have redone these figures (now Figures 6  7). The left columns showing the net difference may appear different, but have the same color table. Since the detachment limited regime only has negative values, the upper range of the color table does not appear.

3.34

Figure 6: if the figure is meant to compare runs with two different rainfall intensities, which intensity was used for the upper and middle figures?

We have completely redone these figures, removed SIMWE's detachment and transport limited cases, and laid out the figures for more direct comparison.

3.35

Software: I tested the model software by installing the latest stable release of GRASS GIS, then going to the GitHub repository for the model's source code. By following the "Basic Instructions" listed there, I was able to install the r.sim.terrain extension and run the example.

3.36

Revised figures attached. A new figure with bivariate scatterplots will be added.

**r.sim.terrain**

input data

↓

determine steady state
or dynamic flow regime

↓

determine model

RUSLE3D — SIMWE — USPED

**RUSLE3D**
- detachment limited regime
- event-based erosivity factor
- flow accumulation
- 3D topographic factor
- detachment rate
- landscape evolution
- gravitional diffusion

_iterate_

**SIMWE**
- variable erosion-deposition regime
- water flow
- erosion-deposition
- landscape evolution
- gravitional diffusion

_iterate_

**USPED**
- transport limited regime
- event-based erosivity factor
- flow accumulation
- 3D topographic factor
- sediment flow at transport capacity
- landscape evolution
- gravitional diffusion

_iterate_

output data

**Fig. 1.** Fig 2

a. Water depth [m] simulated by SIMWE in subwatershed

d. Flow accumulation for RUSLE3D in subwatershed

100 m

b. Sediment flux [kg m$^{-1}$ s$^{-1}$] simulated by SIMWE
in drainage area 1

e. LS3D topographic factor for RUSLE3D
in drainage area 1

c. Erosion and deposition [kg m$^{-2}$ s$^{-1}$] simulated by SIMWE
in drainage area 1

f. Sediment flow with spatially variable landcover
modeled by RUSLE3D [kg m$^{-2}$ s$^{-1}$] in drainage area 1

50 m

**Fig. 2.** Fig 3

[Figure]

a. Subwatershed                                         b. Drainage areas

**Fig. 3.** Fig 4

a. Landcover in 2014

31) Barren Land
41) Deciduous Forest
42) Evergreen Forest
43) Mixed Forest
51) Dwarf Scrub
52) Shrub/Scrub
71) Grassland/Herbaceuous

b. Landforms 2012

1) flat
2) summit
3) ridge
4) shoulder
5) spur
6) slope
7) hollow
8) footslope
9) valley
10) depression

c. Elevation difference between 2012-2016 [m]

1.9
1.1
0.4
-0.4
-1.2

d. Landforms 2016

1) flat
2) summit
3) ridge
4) shoulder
5) spur
6) slope
7) hollow
8) footslope
9) valley
10) depression

50 m

**Fig. 4.** Fig 5

a. Dynamic RUSLE3D net difference [m]

b. Dynamic RUSLE3D landforms

c. Dynamic USPED net difference [m]

d. Dynamic USPED landforms

**Fig. 5.** Fig 6

[Figure]

a. SIMWE simulation net difference [m]          b. SIMWE simulation landforms

**Fig. 6.** Fig 7

---

## Author Response (AR1)

**r.sim.terrain 1.0: a landscape evolution model with dynamic hydrology**

Brendan Alexander Harmon[1], Helena Mitasova[2,3], Anna Petrasova[2,3], and Vaclav Petras[2,3]

[1]Robert Reich School of Landscape Architecture, Louisiana State University, Baton Rouge, Louisiana, USA
[2]Center for Geospatial Analytics, North Carolina State University, Raleigh, North Carolina, USA
[3]Department of Marine, Earth, and Atmospheric Sciences, North Carolina State University, Raleigh, North Carolina, USA

**Correspondence:** Brendan Harmon (baharmon@lsu.edu)

Nota bene: since we have restructured the manuscript, the references to sections, equations, figures, and tables in our responses refer to the revised paper.

**1 Reviewer 1**

**Comment** Although the difference between steady-state and dynamic flow regimes is discussed, the differences between the erosion regimes (e.g. detachment capacity limited, transport capacity limited, erosion-deposition and detachment limited) are less clear. A more thorough discussion of those regimes and their differences would allow for a clearer understanding of the results of the model compared to the typical characteristics associated with these regimes. On P16 L24 to L27, the results of SIMWE were compared to the characteristics typical of the simulated erosion regime. Establishing the characteristics of the erosion regimes earlier, perhaps after the explanation of the flow regimes, would give the reader more clarity regarding what influences these regimes and how the model compares to real-world characteristics.

**Response** We have restructured the paper and now thoroughly discuss soil erosion-deposition regimes in Section 2.1.2 with equations 6-9.

**Comment** Given that the study area has information for 2012 and 2016, one possible improvement is to compare the model results to the observed difference between those two years. Although the results section on P16 compares the modelled characteristics with typical erosion regime characteristics, the comparison to the 2012-2016 data is limited to P16 L23. Adding validation of model results against observed landscape evolution would show the strengths of the model.

**Response** This is a model description paper, rather than a model evaluation paper. While we have added a quantitative comparison of volumetric change to the paper, we plan to conduct a rigorous quantitative evaluation of r.sim.terrain in future work. Because the models are for different erosion regimes, different study sites each with a different dominant regime would be needed to quantitatively assess each model against a relevant baseline. A more accurate, higher frequency of high resolution topographic surveying is also needed, ie. monthly surveys with terrestrial lidar or unmanned aerial systems. We have added

plans for future work to the new Discussion section.

**Comment** Another possible improvement is to more clearly present the limitations of the model in their own section. On P4 L22, the model limitation of not modelling fluvial processes is mentioned. By having a clear limitations section with information about model assumptions, the reader is more informed about the model and how it may affect results.

**Response** We have added a paragraph on the limitations of the model to a new Discussion section.

**Comment** The quality of the figures and the presentation of spatial data is a major issue of the paper. With the exception of Figure 1, many of the figures are too small to be analysed in detail. The legends are pixelated (Figure 4c, 4e, and 4f) or cut off (Figure 5a and 5d). The legends for the landform maps (Figures 5b, 5e, 6b, and 6e) would benefit from the labels presented in Figures 4e and 4f. The colours chosen for the figures could also be improved. For example, Figure 2b shows a landscape with yellow/orange/blue colours but the colour bar only shows a scale of yellow to orange. Using the hillshade layer seems to darken the colours within the gully and the reader is unable to clearly see those colours. The use of a 3D top-down view in Figure 5 makes it difficult to see what is occurring within the gully area where the differences are most important. Some figures are presenting differences (Figure 5c, 5f, 6c) that cannot be visualised clearly because most of them are occurring within the gully area and thus "blocked" by the 3D view and hillshade. Overall, the figures can be improved, especially for visualisation of the key results and differences, and that would contribute to the overall quality of the paper. The differences may be better visualised through 2D top-down view, or 2D cross-sections, or even zooming into the most critical areas of the gully. At the watershed scale and using the current visualisation, the results are difficult to visually interpret and do not supplement the written results well.

**Response** All figures have been redone. They have newer higher resolution legends, scale bars, and north arrows. Figures 3 and 6-9 include details zoomed in on a drainage area (Drainage Area 1) within the subwatershed. These figures are now focused on the main channel of the gully and should show more legible detail. Figure 4b shows the drainage areas. In addition to zooming in on Drainage Area 1, Figures 6-9 are now presented in 2x2 columns and rows, rather than 2x3 columns and rows so that the images are larger and more detail is visible. Selected figures are now 2D rather than 3D maps. We have changed or removed hill shading from select maps to improve their legibility.

**Comment** P3, L22: According to Dabney et al. (2014), RUSLER refers to RUSLE2-Raster which is a distributed form of the Revised Universal Soil Loss Equation Version 2, which is normally referred to as RUSLE2. The paper is referring to the Revised Universal Soil Loss Equation Version 2 when it is using the RUSLE2-Raster acronym. Please clarify if the paper is referring to RUSLER or RUSLE2.

**Response** We have replaced this with: "Gully erosion has been simulated with RUSLE2-Raster (RUSLER) in conjunction with the Ephemeral Gully Erosion Estimator (EphGEE) (Dabney et al., 2014)."

**Comment** P6, L11 to P7, L6: This paragraph would be better presented in a table or a flowchart showing how the model switches erosion regimes based on rainfall intensity.

**Response** To more clearly present how soil erosion regimes are handled in this model we have added Section 2.1.2 Erosion-Deposition Regimes. We have also removed the detachment limited and transport limited cases for SIMWE to avoid unnecessary complexity in the paper and results.

**Comment** P14, L11 to L13: Additional detail about how the information about K-factor, C-factors, Manning's, and runoff rates were derived would be useful for those who wish to apply the model in their study area.

**Response** We have added links to detailed instructions for deriving these maps in the tutorial. We have also added a link to our data log with a complete record of the commands used to process the sample data.

**Comment** P3, L17: Since LIDAR is an acronym for Light Detection and Ranging, mentions of LIDAR should be in capitals and the first instance should have the accompanying meaning of LIDAR.

**Response** We have followed the recommendation in the paper "Let's agree on the casing of lidar" ( Deering & Stoker, 2014), which shows that 65% of literature uses lidar (including USGS), while 17% use LIDAR and 14% use LiDAR. Their reasoning is that lidar is the most common usage, the original usage, and the usage recommended by style manuals However, the latest issues of ISPRS journals use LiDAR, so we will defer to the journal editor on what their standard should be. For our part we prefer to follow USGS usage of lidar.

Deering, Carol & Stoker, Jason. (2014). Let's agree on the casing of lidar. Lidar Magazine. 4. 48-51. http://lidarmag.com/wp-content/uploads/PDF/LiDARNewsMagazine_DeeringStoker-CasingOfLiDAR_Vol4No6.pdf

**Comment** P4, L28 and similar headings: For these headings, referee suggests formatting as follows "Simulation of Water Erosion Model (SIMWE)" and only using the acronym on the following line.

**Response** Reformatted as recommended.

**Comment** P6, Table 1: Citation of "(Dennis C. Flanagan et al., 2013)" should just be "(Flanagan et al., 2013)".

**Response** Citation fixed as recommended.

**Comment** P10, L15: The addition of "(a)" after "The upslope contributing area per unit width" would allow for a clearer connection to Equation 12.

**Response** Variable added as recommended.

**Comment** P13, L14 and P14, L1: Scientific names should be italicised.

**Response** Scientific names have been italicized.

**2 Reviewer 2**

**Comment** The text has detailed descriptions of the underlying theory behind many components of the overall simulation model (section 2) but does not indicate how these relate to one another. While there are many colorful figures of landscape evolution results, there is no representation of the model itself (e.g., UML or other flow-chart-like visualization) – nor is there much in the way of narrative description of how the model actually works to combine the described elements.

**Response** We have added a conceptual diagram of the model as Figure 2.

**Comment** Section 3.2 describes several experimental runs with the simulation model. These experiments are summarized in Table 3. However, we have no quantitative information about the results of the experiments. There is no information about whether each experiment was run only once or repeated–or whether repetition is needed or not because each parameter setting produces or does not produce only one outcome. Although the simulation is situated in a realistic setting based on digital data from Ft. Bragg, NC, the rationale for some of the parameterizations are not given, particularly the important rainfall settings. These might be completely reasonable, but the authors should indicate if these are based on empirical rainfall data or have another basis (e.g., extreme values to test the model sensitivity).

**Response** Repeat runs are not needed because RUSLE and USPED are empirical, non-stochastic models and produce only one outcome. SIMWE path sampling method includes stochastic component for solving the continuity equations but this relates to the accuracy of the solution, e.g., a high number of walkers reduces the numerical error associated with the path sampling solution. We have added a discussion of this to the paragraph on limitations. We have added more information about the parameters of the simulations in subsections 3.1 and 3.2 on the study site and simulations. This includes a link to detailed instructions for

the sample data and a discussion of the design storms and their rationale.

**Comment** The authors have collected detailed, time series, LiDAR and orthophoto data from the test area represented in the simulation. But they make no attempt to compare the simulation experiments with these data in any quantitative sense. Rather they give only brief, subjective assessments of model behavior. It would seem rather easy to compare the model with the empirical data to see which experiments are better or worse fits and in what ways.

**Response** We consider this manuscript a model description paper, rather than a model evaluation paper. While we have added a quantitative comparison of volumetric change to the paper, we plan to conduct a rigorous quantitative evaluation of r.sim.terrain in future work. Because the models are for different erosion regimes, different study sites with a different dominant regimes may be needed to quantitatively assess each model against a relevant baseline. A higher frequency of high resolution topographic surveying would also help, ie. monthly surveys with terrestrial lidar or unmanned aerial systems. We have added plans for future work to the conclusion.

**Comment** Finally, I downloaded and installed r.sim.terrain into GRASS and tried to run it. I strongly commend the authors for making the model code and test data available. This is critically important for research based on modeling like this paper. I installed r.sim.terrain into the most current version of GRASS according to the directions in the manuscript (i.e., using g.extension). Unfortunately I ran into several problems that made testing the the model impossible. First, there is a link to the test dataset in the model online help, but this link does not work. Using information in the paper, I was able to go to the GitHub site and poke around until I found the test data set and installed it into my GRASS data directory. I then followed the steps in the tutorial to simply see how it ran–thinking to compare the different overland flow and erosion/deposition methods, and time series vs. event. The command given to test the model failed initially because it was missing the rather critical "elevation" argument. So I added that. Then it started but rapidly bombed with errors related to the time series part. I copy these below. So I never did get it to run.

**Response** We have updated the documentation both on the manual page and the GitHub repository with the tutorial. The reviewer's run of the model failed due to incomplete documentation rather than bugs. The input elevation raster must be in the current mapset (for registration in the temporal database), so it should be copied from the PERMANENT mapset to the current working mapset before the model is run. We have added a section with basic instructions to the manual page (https://grass. osgeo.org/grass76/manuals/addons/r.sim.terrain.html) and to the repository readme (https://github.com/baharmon/landscape_ evolution). We have also written a longer tutorial (https://github.com/baharmon/landscape_evolution/blob/master/tutorial.md) that details running each model with instructions and examples for RUSLE3D, USPED, and SIMWE.

**Comment** However, the authors need to do a better job of explaining how the model works and not just the conceptual components included in the model. They also need to provide more information about the four experiments performed and their

parameter settings. They need to provide some quantitative evaluation of the model results, including comparison with the empirical data they have collected. Finally, they need to fix some probably minor but annoying bugs in the code available for evaluation.

**Response** We have added a conceptual diagram of the model (Figure 2) and rewritten and expanded subsection 3.2 on the simulations and their parameters. We have added quantitative evaluation and comparison of the models and explained its limitations. We, however, would like to reiterate that this is a model description rather than evaluation paper. Finally, while there were no bugs in the code, we have fixed and expanded the documentation, which was incomplete.

**3 Reviewer 3**

**Comment** 1/title - This is a total quibble, and please feel free to ignore it, but my first reaction to 'dynamic landscape evolution model' was to ask (rhetorically) 'is there any other kind'? Consider 'landscape evolution model with dynamic hydrology' as an alternative (admittedly a less pithy one).

**Response** Titled changed to "r.sim.terrain: a landscape evolution model with dynamic hydrology" as recommended.

**Comment** 1/5 'steady state or dynamic model' could be read as implying that the entire model is steady state, not just the surface water flow rates. Suggest re-wording: 'using either a steady state or dynamic representation of overland flow ...'

**Response** Reworded as recommended.

**Comment** 2/2 I agree with the sentiment, but suggest rewording to 'a landscape evolution model that includes time-evolving surface water discharge', to avoid confusion over which aspect of the model is dynamic.

**Response** Replaced with "A landscape evolution model with dynamic water and sediment flow..."

**Comment** 3/8 The phrase 'until water flow reaches steady state' suggests that the positive feedback (presumably between deepening/widening and attraction of more surface water flow) stops at this point. I don't think that is necessarily true; you could have a feedback between morphology and flow under steady runoff too.

**Response** An excellent point. Revised simply by cutting "until water flow reaches steady state."

**Response** 3/11 Please explain what is meant by erosion-deposition regime.

**Comment** We have added a new Section 2.1.2 describing erosion-deposition regimes. This includes equations 7-11.

5 **Response** 3/14 Detachment vs transport capacity: this sounds backwards...

**Comment** Detachment capacity and transport capacity are now more clearly explained in Section 2.1.2 Erosion-Deposition Regimes.

10 **Response** 3/18-19 There are plenty of other papers that could be cited here, in which one or more of the listed methods was used to study gully erosion. (For example, here's a review paper that cites some TLS applications to gully erosion: Telling, J., Lyda, A., Hartzell, P., & Glennie, C. (2017). Review of Earth science research using terrestrial laser scanning. Earth-Science Reviews, 169, 35-68.)

15 **Response** This paragraph was just meant to be a brief overview of methods, not a comprehensive review, but we have added many more references to the introduction.

**Comment** Figure 2: please give location in caption. Also, numbers on color bars and scale bar are barely legible.

20 **Response** We added the location to the caption and have redone the legends and scale bars at higher resolution.

**Comment** 6/2 typo.

**Response** The typo has been fixed.

**Comment** 6/2 I guess 'partial derivatives of the topography' means a numerical approximation of the derivative of the elevation field with respect to the two cardinal grid directions. Recommend more precision in wording here.

**Response** We have more clearly phrased this, explained briefly how it is computed, and added a reference to the chapter Geo-
30 morphometry in GRASS GIS (Hofierka et al., 2009) that explains the math and implementation.

**Comment** 6/9 'steady state dynamics' - I think I understand what you mean here, but the phrase itself is awkward (it is self-contradictory).

**Response** Replaced with: "r.sim.terrain simulates unsteady-state flow regimes when the landscape evolution time step is less than the travel time for a drop of water or a particle of sediment to cross the landscape, e.g. when the time step is less than the time to concentration for the modeled watershed. With longer landscape evolution time steps the model simulates a steady state regime."

**Comment** Table 2: this is only a partial list of codes that have been published in, say, the last ten years. Why choose these particular ones?

**Response** We removed the table and instead list or briefly discuss these landscape evolution models and others in the body of the Introduction.

**Comment** Table 2: Be careful about giving the spatial scale for these models. At least some of these codes have been used and published at a variety of different spatial scales, from say the size of a rilled hillslope to that of a small country; and in some cases (e.g., SIBERIA) is sometimes presented in a dimensionless mode in which no spatial scale at all is given or implied. As to temporal scale, I thought that at least some of these can also be run in 'event' mode. Also, my understanding is that Landlab is not itself a model, but rather is a programming library that contains components that can be used to build various types of model, including landscape evolution. That said, people seem to have built landscape evolution models using Landlab (the Landlab website lists some of these). Maybe it would make sense to label this entry as 'Landlab-built erosion models' or something like that.

**Response** We removed the table and instead list or briefly discuss these libraries and landscape evolution models in the text.

**Comment** Section 2.1 generally: I like the way that this is carefully organised into sub-sections. However, the order of presentation confused me. Often, authors presenting a set of governing equations will start with the high-level conservation law(s), and then define each term more precisely. As noted below, there's an opportunity to do this at least partly in subsection 2.1.1.

**Response** We agree and we have started this description with the general equation for change in elevation (continuous form of eq. 7) followed by general equation for $d_s$. See e.g. eq. (9) in Mitasova et al. 2005 ($D(r,t)$ is our $d_s$).

**Comment** Equation 2: it would be helpful to give some context and referencing. I think this idea comes from Foster and Meyer (1972), right? If I remember correctly, their key assumption was that the ratio of transport rate to transport capacity, plus the ratio of detachment rate to detachment capacity, sum to unity. Assuming I did the math right, this leads to a first-order reaction-like equation: $dz/dt = ds = sigma(qs - Tc)$ I recommend presenting it this way here in section 2.1.1 (in addition to the definition given in eq 2), because this relates transport and detachment to the rate of change of elevation, and motivates the need for definitions for qs, Tc, and Dc. Note that there seems to be a problem with units in one of the factors in eq 2: if Tc and

Dc had the same units (as is listed), then sigma would be dimensionless. I suspect Dc is actually in $kgm^{-2}s^{-1}$ (detached mass per unit area per time).

**Response** We have added a more complete explanation of the Foster and Meyer relationship and the related parameters and fixed the units for $D_c$ (eq. 12-13 in Mitasova et al. 2005).

**Comment** Equation 4: symbol v is used without being introduced. Presumably it is the depth averaged flow velocity vector in (x,y). Either define v or use q (which you've defined already).

**Response** We modified the equation according to the reviewer's suggestion.

**Comment** Also, whereas the paper is premised on the value of having a dynamic representation of surface-water hydrology (which eq 3 is), equation 4 is actually a steady solution, is it not? If the model indeed uses a fully time-varying flow model, the equations presented in this sub-section should show this. In addition, it would be helpful to provide a reference for this form of the diffusion-wave approximation (could be to a hydrology text that gives the derivation and assumptions).

**Response** The in-depth explanation of mathematical foundations for the shallow water flow simulation has been addressed in several previously published papers, for example Mitasova et al. 2005, and we tried to avoid repeating text presented there, unfortunately this makes some of the concepts and reasoning behind the methods less clear. Therefore, we have rewritten the entire section 2 including more detailed explanation of the method and more specific references.

**Comment** "Please give units of epsilon."

**Response** Units for epsilon are now given in the text.

**Comment** 8/10 suggest specifying '...density in the water column', so it is clear that this is a mass concentration rather than a bulk density of resting sediment.

**Response** Changed as recommended by reviewer.

**Comment** 8/15 "steady state sediment flow with diffusion" - I'm confused by this. The equation is time-dependent, so how is it steady state? And the definition of qs above is advective, not diffusive.

**Response** We added the equation with diffusion term used in the path sampling solution, see e.g. Mitasova et al. 2005 eq. 16.

**Comment** 8/17 So we need a definition for ds, which as suggested above, you could provide in section 2.1.1.

**Response** Agreed. We rewrote section 2.1 and included a definition of ds.

**Comment** 8/23 In the previous equation, you used a continuum formulation, whereas here you're giving a discretized-in-time form. Please be consistent. I suggest sticking with continuum forms, because these don't require you to make any statements about numerical approximation. And in fact, as noted above, I recommend putting equation 7 in section 2.1.1.

**Response** We have rewritten the equation into continuum form and moved section 2.1.5 to section 2.1.1 following the reviewer's recommendation.

**Comment** Equation 8: this equation is not dimensionally consistent. If you write it in continuum form, $dz/dt = -(1/rho)qs$ you have m/s on the left and m2/s on the right. I'm also not convinced that the equation expresses the idea you want. I'm guessing that a detachment-limited regime would look more like $dz/dt = -(1/rho)Dc$. Then it becomes a question of what is your detachment capacity law? You've already introduced detachment capacity in $Dc = sigmaTc$ (eq 2). In order to close the equations, you need either a definition of Dc or Tc. Presumably these depend in some fashion on water discharge or velocity or boundary shear stress. Please specify (or, if I have misunderstood, explain why the equation set given is sufficient to describe the SIMWE module). Actually, after reading farther in the manuscript, I think the idea is that the RUSLE equation can be used for Dc in detachment-limited mode. If that's correct, then say something to the effect that the definition of Dc will be given in section so-and-so, and then use the symbol Dc in that section. Regarding the role of qs, I suspect what you're after is the notion that qs is the upstream/upslope integral of ds, is that right? If so, it would be helpful to present the math.

**Response** You are correct, using $q_s$ here was an oversight. Eq. 8 is not really needed, because $d_s$ in DLC is erosion rate given by eq 13 which is detachment rate (soil loss) not sediment flow. We have removed this equation.

**Comment** 9/21 please give the functional form of this relationship.

**Response** The equation for this relationship is given in the section 2.2.3.

**Comment** 10/2 I get that there's a long tradition of practical empiricism in soil-erosion research. But what about pushing ever so gently back on it by presenting equation 10 in a slightly less brutally ugly form? Something like: $er/e_ref = 1 - aexp(-ir/i_ref)$ where $e_ref$ is reference energy equal to ... and $i_ref$ is reference rainfall intensity equal to.

**Response** Thank you for the suggestion. We have modified the equation accordingly.

**Comment** "10/7 shouldn't this be rainfall depth rather than volume? Equation 11: again the units seem to be off here (apart from the oddity of having an 'index' that has [weird] units). I get the right side as being: $MJha^{-1}mm^{-1}xmmxs = MJha^{-1}s$?"

**Response** We have revised this equation, checked the units, and introduced it with Equation 2 from Panagos et al. 2015 from which it is derived.

**Comment** 11/3 the subsection is called 'Sediment flow' but it reads like an erosion rate. Though I guess it works given that you're defining it as mass flow per time per area.

**Response** You are right – it is an erosion rate (soil loss: mass per area per time). In this paper we were using the term sediment flow for sediment flow per unit width mass per length per time). We have retitled this subsection as Detachment Limited Erosion Rate and replaced sediment flow with erosion rate throughout the paper for this case.

**Comment** Equation 13: again I'm struggling with units. I get: $(MJmmha^{-1}hr^{-1})x(tonhahrha^{-1}MJ^{-1}mm^{-1}) = (ha^{-1})x(ton)$ which are not the units given for E.

**Response** We have revised the units for E and R.

**Comment** 11/16-17 it's not clear to me how these equations relate. Maybe you mean that the definition of E in equation 13 is the SAME AS ds (or -ds) for transport-limited conditions, and Dc for detachment-limited conditions? In that case, it might suffice to simply call equation 13 the definition of Dc. You could then give the definition of Tc as sigma = Dc/Tc ==> Tc = Dc/sigma (Note: it would be more intuitive to think in terms of a length scale, L = 1 / sigma, which is then the characteristic distance over which steady, uniform overland flow reaches its carrying capacity on a planar slope).

**Response** We have renamed this section and revised the text following the reviewer suggestion. The equation 8 was removed (see the related answer above).

**Comment** 11/29 not clear to me what 'topographic component of overland flow' means.

**Response** We revised section 2.4.1 "Topographic sediment transport factor".

**Comment** Equation 15: is $T$ the same as $T_c$? Also, again, I'm not sure the units are correct here, please check, and correct if necessary.

**Response** Yes, $T$ is $T_c$. We have unified the symbols and checked the units.

**Comment** Figures 5 and 6: why the different color schemes in two of the three comparisons (top and bottom rows)?

**Response** We have redone these figures (now 7-9). The subfigures showing the net difference may appear different, but have the same color table. Since the detachment limited regime only has negative values, the upper range of the color table does not appear.

**Comment** Figure 6: if the figure is meant to compare runs with two different rainfall intensities, which intensity was used for the upper and middle figures?

**Response** We have completely redone these figures, removed SIMWE's detachment and transport limited cases, and laid out the figures for more direct comparison.

**Comment** Software: I tested the model software by installing the latest stable release of GRASS GIS, then going to the GitHub repository for the model's source code. By following the "Basic Instructions" listed there, I was able to install the r.sim.terrain extension and run the example.

**r.sim.terrain 1.0: a [..*]landscape evolution model with dynamic hydrology**

Brendan Alexander Harmon[1], Helena Mitasova[2,3], Anna Petrasova[2,3], and Vaclav Petras[2,3]

[1]Robert Reich School of Landscape Architecture, Louisiana State University, Baton Rouge, Louisiana, USA
[2]Center for Geospatial Analytics, North Carolina State University, Raleigh, North Carolina, USA
[3]Department of Marine, Earth, and Atmospheric Sciences, North Carolina State University, Raleigh, North Carolina, USA

**Correspondence:** Brendan Harmon (baharmon@lsu.edu)

**Abstract.** While there are numerical landscape evolution models that simulate how steady state flows of water and sediment reshape topography over long periods of time, r.sim.terrain is the first to simulate short-term topographic change for both steady state and dynamic flow regimes across a range of spatial scales. This free and open source, GIS-based topographic evolution model uses empirical models for soil erosion [..²]and a physics-based model for shallow overland water flow and soil erosion
5 [..³]to compute short-term topographic change. This [..⁴]model uses either a steady state or [..⁵]unsteady representation of overland flow to simulate how overland sediment mass flows reshape topography for a range of hydrologic soil erosion regimes based on topographic, land cover, soil, and rainfall parameters. As demonstrated by a case study for Patterson Branch subwatershed on the Fort Bragg military installation in North Carolina, r.sim.terrain [..⁶]simulates the development of fine-scale morphological features including ephemeral gullies, rills, and hillslopes. Applications include land management, erosion
10 control, landscape planning, and landscape restoration.

*Copyright statement.* ...

**1 Introduction**

Landscape evolution models represent how the surface of the earth changes over time in response to physical processes. Most studies of landscape evolution have been descriptive, but a number of numerical landscape evolution models have been
15 developed that simulate elevational change over time [..¹⁵](Tucker and Hancock, 2010; Temme et al., 2013). Numerical landscape evolution models such as the Geomorphic - Orogenic Landscape Evolution Model (GOLEM) (Tucker and Slinger-
* * *
*removed: dynamic
²removed: at watershed to regional scales
³removed: at subwatershed scales
⁴removed: either
⁵removed: dynamic model simulates
⁶removed: can realistically simulate
¹⁵removed: (Temme et al., 2013)

[Figure]

a.                                                                                   b.

[revised manuscript text omitted]

15 as the divergence of sediment flow (Tucker et al., 2001):

$$\frac{\partial z}{\partial t} = (-\nabla \cdot \mathbf{q_s} i) \ \rho_s^{-1} = d_s \ \rho_s^{-1} \tag{2}$$
* * *
[50]removed: This

[52]removed: Simulation of water erosion model

[Figure]

**Figure 3.** Water and sediment flows modeled by (a & c) SIMWE and (b & d) RUSLE3D with spatially variable landcover for a (a & b) subwatershed and (c & d) drainage area of Patterson Branch, Fort Bragg, NC. (a) Water depth [m] simulated by SIMWE for a 10 min event with 50 $\mathrm{mm\,hr^{-1}}$ in the subwatershed. (b) Flow accumulation for RUSLE3D in the subwatershed. (c) Erosion and deposition $[\mathrm{kg\,m^{-2}\,s^{-1}}]$ simulated by SIMWE in drainage area 1. (d) Erosion $[\mathrm{kg\,m^{-2}\,s^{-1}}]$ modeled by RUSLE3D [..[51] ]in drainage area 1.

where:

$z$ is elevation [m]

$t$ is time [s]

$\mathbf{q_s}$ is sediment flow per unit width (vector) [$\mathrm{kg\,m^{-1}\,s^{-1}}$]

$d_s$ is the net erosion-deposition rate [$\mathrm{kg\,m^{-2}\,s^{-1}}$]

$\rho_s$ is sediment mass density [$\mathrm{kg\,m^{-3}}$].

The net erosion-deposition rate $d_s$ driven by overland flow in r.sim.terrain is estimated at different levels of complexity based on the simulation mode selected by the user. Gravitational diffusion is then applied to the changed topography to simulate the smoothing effects of localized soil transport between rainfall events. The change in elevation due to gravitational diffusion is a function of the sediment mass density, the diffusion coefficient, and Laplacian of the elevation (Thaxton, 2004):

$$\frac{\partial z}{\partial t} = \rho_s^{-1}\,\varepsilon_g\,\nabla^2 z \tag{3}$$

where $\varepsilon_g$ is the diffusion coefficient [$\mathrm{kg\,m^2\,s^{-1}}$].

The discrete implementation follows Thaxton (2004):

$$z_{t+\Delta t_1} = z_t + \Delta z_s \tag{4}$$

$$z_{t+\Delta t_1 + \Delta t_2} = z_{t+\Delta t_1} + \Delta z_g \tag{5}$$

where:

$\Delta z_s$ is elevation change caused by net erosion or deposition [m] (Eq. 2)

$\Delta z_g$ is the diffusion driven elevation change [m] (Eq. 3)

$\Delta t_1$ is the time interval during a storm event [s]

$\Delta t_2$ is the time interval between events when gravitational diffusion changes the elevation surface [s].

**2.1.2 Erosion-deposition regimes**

[revised manuscript text omitted]

**2.2.1 [..[62] ]**

[..[63] ]

$$\sigma = \frac{D_c}{T_c}$$
* * *
[53]removed: – the Simulation of Water Erosion model –

[54]removed: and momentum

[55]removed: determines the soil erosion regime, simulates water and sediment flows, and then evolves the topography. In an variable erosion-deposition regime the model computes the

[56]removed: topography,

[57]removed: and

[58]removed: ,

[59]removed: The same process is used in a transport capacity limited regime, except that the topography is evolved based on the transport limited erosion-deposition rate and gravitational diffusion. In a detachment capacity limited regime the model instead computes the

[60]removed: topography, simulates shallow water flow and sediment flow, and then evolves the topography based on the sediment flow rate and gravitational diffusion. The model simulates dynamic landscape evolution when the

[61]removed: steady state dynamics.

[62]removed: Erosion regime

[63]removed: This model can switch erosion regimes at each time step based on the rainfall intensity ($i_r$) and the balance of the sediment detachment capacity ($D_c$) and the sediment transport capacity ($T_c$) represented by the first order reaction term $\sigma$, which depends on soil and landcover properties. The detachment capacity is the maximum potential detachment rate by overland flow, while the sediment transport capacity is the maximum potential sediment flow rate. When rainfall intensity is very high ($i_r \geq 60\,\mathrm{mm\,hr^{-1}}$) or $\sigma$ is low ($\sigma \leq 0.01\mathrm{m^{-1}}$), then the regimeis detachment capacity limited. When rainfall intensity is not very high ($i_r < 60\,\mathrm{mm\,hr^{-1}}$) and $\sigma$ is high ($\sigma \geq 100\mathrm{m^{-1}}$), then the regime is transport capacity limited. When rainfall intensity is not very high ($i_r < 60\,\mathrm{mm\,hr^{-1}}$) and $\sigma$ is neither high nor low ($0.01\mathrm{m^{-1}} < \sigma < 100\mathrm{m^{-1}}$), then there is an variable erosion-deposition regime.

[..64 ]

[..64 ]
[..65 ][[..66 ]]
5   [..67 ][[..68 ]]
[..69 ][[..70 ]]
a steady state regime.

**2.2.1   Shallow water flow**

The SIMWE model simulates shallow overland water flow controlled by spatially variable topographic, soil, landcover, and
10   rainfall parameters by solving the [..71 ]water flow continuity equation using a Green's function Monte Carlo path sampling
method[..72 ]:

$$\frac{\partial h}{\partial t} = i_e - \nabla\, q$$

[..73 ]
[..74 ][[..75 ]]
15   [..76 ][[..77 ]]
[..78 ][[..79 ]]
[..80 ][[..81 ]]
[..82 ]
[..83 ]
20   [..84 ][[..85 ]][..86 ]
* * *
[64]removed: where:

[65]removed:   $\sigma$ is a first order reaction term

[66]removed: m$^{-1}$

[67]removed:   $D_c$ is the sediment detachment capacity

[68]removed: kg m$^{-1}s^{-1}$

[69]removed:   $T_c$ is the sediment transport capacity

[70]removed: kg m$^{-1}s^{-1}$

[71]removed: continuity and momentum equations for steady state water flow with a

[72]removed: (Fig. 3a ). Shallow water flow $q$ can be approximated by the bivariate form of the St. Venant equation:

[73]removed: where:

[74]removed:   $x, y$ is the position

[75]removed: m

[76]removed:   $t$ is the time

[77]removed: s

[78]removed:   $h$ is the depth of overland flow

[79]removed: m

[80]removed:   $i_e$ is the rainfall excess

[81]removed: m s$^{-1}$

[82]removed:   (i.e. rainfall intensity $-$ infiltration $-$ vegetation intercept)

[83]removed:   $\nabla$ is the divergence of the flow vector field

[84]removed:   $q$ is the water flow per unit width

[85]removed: m$^2$ s$^{-1}$

[86]removed: .

$$\nabla \cdot \boldsymbol{q} = i_e \tag{10}$$

where:

$i_e$ is the rainfall excess rate $[\mathrm{m\,s^{-1}}]$ (i.e. rainfall intensity $-$ infiltration $-$ vegetation intercept)

5     $\boldsymbol{q}$ is the water flow per unit width (vector) $[\mathrm{m^2\,s^{-1}}]$.

[..[87] ]The path sampling method solves the continuity equation through the accumulation of the evolving source over the given time period. This accumulation process can be interpreted as an approximation of a dynamic solution with diffusive wave effects incorporated by adding a diffusion term proportional to $\nabla^2[h^{5/3}]$ in the solution:

$$-\frac{\varepsilon_w}{2}\nabla^2\, h^{5/3} + \nabla \cdot \boldsymbol{q} = i_e \tag{11}$$

10   where:

$\varepsilon_w$ is a spatially variable diffusion coefficient $[\mathrm{m^{4/3}\,s^{-1}}]$.

See Mitasova et al. (2004) for more details on this equation and its numerical solution. The solution assumes that water flow velocity is largely controlled by the slope of the terrain and surface roughness and that its change at a given location during the simulated event is negligible. The water depth $h$ at time $\tau$ during the simulated rainfall event is computed as

15   a function of particle (walkers) density at each grid cell. The initial number of particles per grid cell is proportional to the rainfall excess rate $i_e$ (source). Particles are then routed across the landscape by finding a new position for each walker at time $\tau + \Delta\tau$:

$$\boldsymbol{r}_\mathsf{m}^\mathsf{new} = \boldsymbol{r}_\mathsf{m} + \Delta\tau \boldsymbol{v} + \boldsymbol{g} \tag{12}$$

where:

20     $\boldsymbol{r} = (x, y)$ is the $m^{th}$ walker position [m]

$\Delta\tau$ is the particle routing time step [s]

$\boldsymbol{g}$ is a random vector with gaussian components with variance $\Delta\tau$ [m]

$\boldsymbol{v}$ is the water flow velocity vector $[\mathrm{m\,s^{-1}}]$ whose magnitude is computed with the Manning's equation $v = n^{-1}\, h^{2/3}\, s^{1/2}$ where $n$ is the Manning's coefficient $[\mathrm{s\,m^{-1/3}}]$ and $s$ is slope.

25   The mathematical background of the method, including the incorporation of approximate momentum through an increased diffusion rate in the prevailing direction of flow, is presented by Mitas and Mitasova (1998) and Mitasova et al. (2004).

**2.2.2   Sediment flow and net erosion-deposition**

The SIMWE model simulates the sediment flow over complex topography with spatially variable overland flow, soil, and landcover properties by solving the sediment flow continuity equation using a Green's function Monte Carlo path sampling
* * *
[87] removed: Diffusive wave effects can be approximated so that water can flow through depressions by integrating a diffusion term $\propto \nabla^2[h^{5/3}]$ into the solution of the continuity and momentum equations for steady state water flow . This equation is solved

method.

$$-\frac{\varepsilon}{2}\nabla^2[h^{5/3}] + \nabla\,[h\,v] = i_e$$

5    [..[88] ]

[..[89] ]

**2.2.3  [..[90] ]**

[..[91] ]Steady state sediment flow $q_s$ is approximated by the bivariate continuity equation, which relates the change in
10  sediment flow rate [..[92] ]to effective sources and sinks:

$$\nabla \cdot q_s = \text{sources} - \text{sinks} = d_s \tag{13}$$

The sediment-flow rate $q_s$ is a function of water flow and sediment concentration (Mitas and Mitasova, 1998)[..[93] ]

$$q_s = \rho_s\,q$$

[..[94] ]
15  [..[95] ][[..[96] ]]
[..[97] ][[..[98] ]][..[99] ]

**2.2.3  [..[100] ]**

[..[101] ]:

$$d_s = \frac{\partial[\rho_s\,h]}{\partial t} + \nabla\,q_s$$
* * *
[88]removed: where:

[89]removed:  $\varepsilon$ is a spatially variable diffusion coefficient.

[90]removed: Sediment flow

[91]removed: In SIMWE the

[92]removed: $q_s$ is estimated as

[93]removed: (Fig. 3b):

[94]removed: where:

[95]removed:  $q_s$ is the sediment flow rate per unit width

[96]removed: $\text{kg m}^{-1}\,\text{s}^{-1}$

[97]removed:  $\rho_s$ is sediment mass density

[98]removed: $\text{kg m}^{-3}$

[99]removed: .

[100]removed: Erosion-deposition

[101]removed: In SIMWE the net erosion-deposition rate is estimated using the bivariate form of sediment continuity equation to model sediment storage and flow based on effective sources and sinks (Fig. 3c). Net erosion-deposition $d_s$ – the difference between sources and sinks – is approximated by the steady state sediment flow equationwith diffusion

[..102 ]

[..[102] ]

[..[102] ]

[..[103] ][[..[104] ]][..[105] ]

$$q_s = \rho_s \, c \, q = \rho_s \, c \, h \, v = \varrho \, v \tag{14}$$

where:

$\rho_s$ is sediment mass density in the water column $[\mathrm{kg\,m^{-3}}]$

$c$ is sediment concentration $[\mathrm{particle\,m^{-3}}]$

10    $\varrho = \rho_s \, c \, h$ is the mass of sediment transported by water per unit area $[\mathrm{kg\,m^{-2}}]$.

**2.2.3   [..[106] ]**

The sediment flow equation (13), like the water flow equation, has been rewritten to include a small diffusion term that is proportional to the mass of water-carried sediment per unit area $\nabla^2 \varrho$ (Mitas and Mitasova, 1998):

$$-\frac{\varepsilon_s}{2}\nabla^2\varrho + \nabla \cdot [\varrho v] + \varrho\,\frac{D_c}{T_c}\,|\mathbf{v}| = D_c \tag{15}$$

15   where:

$\varepsilon_s$ is the diffusion constant $[\mathrm{m^2\,s^{-1}}]$.

[..[107] ]On the left hand side of equation 15, the first term describes local diffusion, the second term is drift driven by the water flow, and the [..[108] ] [..[109] ]

20    third term represents a velocity dependent 'potential' acting on the mass of transported sediment. The initial number of particles per grid cell is proportional to the soil detachment capacity $D_c$ (source). The particles are then routed across the landscape by finding a new position for each the walker at time $\tau + \Delta\tau$:

$$r_m^{\text{new}} = r_m + \Delta\tau v + g \tag{16}$$

while the updated weight is:

$$w_m^{new} = w_m \exp[-\Delta\tau(u(\boldsymbol{r}_m^{new}) + u(\boldsymbol{r}_m))/2] \tag{17}$$
* * *
[102]removed: where:

[103]removed:   $d_s$ is net erosion-deposition

[104]removed: $\mathrm{kg\,m^{-2}\,s^{-1}}$

[105]removed: .

[106]removed: Landscape evolution

[107]removed: The simulated change in elevation $\Delta z$ due to water erosion and deposition is a function of the change in time, the net erosion-deposition rate

[108]removed: sediment mass density (Mitasova et al., 2013):

[109]removed:

$\Delta z = \Delta t \, d_s \, \rho_s^{-1}$

5    where:

$$u = D_c/T_c \, |\boldsymbol{v}|.$$

[..[110] ]The sediment flow rate [..[111] ]

$$\Delta z = \Delta t \, q_s \, \varrho_s^{-1}$$

[..[112] ]

10    [..[113] ][[..[114] ]][..[115] ]

is computed as the product of weighted particle densities and the unit vector in the direction of flow $q_s = \varrho \, s_0$. Then net erosion-deposition $d_s$ is computed as the divergence of sediment flow using equation (13).

    SIMWE estimates the detachment capacity $D_c$ and the sediment-transport capacity $T_c$ as functions of shear stress and stream power respectively. Specifically, the detachment capacity is:

15    $D_c = K_d\big(\gamma - \gamma_0\big)^b$                                                       (18)

where:

   $K_d$ is the effective erodibility (detachment-capacity coefficient) $[\mathrm{s\,m^{-1}}]$ for $b = 1$

   $\gamma = \rho_w \, g \, h \, \sin\beta$ is shear stress $[\mathrm{Pa} = \mathrm{kg\,m^{-2}}]$

   $\rho_w$ is the mass density of water $[\mathrm{kg\,m^{-3}}]$

20    $g$ is gravitational acceleration $[\mathrm{m\,s^{-2}}]$

   $\gamma_0$ is critical shear stress [Pa]

   $b$ is an empirical exponent.

[..[116] ]Shear stress $\gamma$ is computed as a function of water depth $h$ estimated by r.sim.water and the surface slope angle $\beta$ [°]. Sediment-transport capacity is computed as a function of [..[117] ]unit stream power $\omega$ (Moore and Burch, 1986):

25    $T_c = K_s\omega = K_s \, \gamma \, |\mathbf{v}| = K_s \, n^{-1} \, g_w \, h^m \, (\sin\beta)^p$                                     (19)

where:

   $K_s$ is the effective sediment-transport capacity coefficient [s]

   $m$ and $p$ are empirical exponents.

    This model can simulate erosion regimes from prevailing detachment limited conditions when $T_c >> D_c$ to prevailing transport capacity limited conditions when $D_c >> T_c$ and the erosion-deposition patterns between these conditions.
* * *
[110] removed: In a detachment limited erosion regime the simulated change in elevation $\Delta z$ is a function of the change in time , the

[111] removed: , and the mass of water carried sediment per unit area (Mitasova et al., 2013):

[112] removed: where:

[113] removed:    $\varrho_s$ is the mass of sediment per unit area

[114] removed: kg m$^{-2}$

[115] removed: .

[116] removed: Gravitational diffusion is then applied to the evolved topography to simulate the settling of sediment particles.The simulated change in elevation $\Delta z$ due to gravitational diffusion is

[117] removed: the change in time , the sediment mass density, the gravitational diffusion coefficient, and topographic divergence –

[revised manuscript text omitted]

where:

$LS_{3D}$ is the dimensionless topographic [..$^{149}$ ]factor

$a$ is upslope contributing area per unit width [m]

$a_0$ is the length of the standard USLE plot [22.1 m]

$\beta$ is the angle of the slope [°]

5   $m$ is an empirical coefficient

$n$ is an empirical coefficient

$\beta_0$ is the slope of the standard USLE plot [0.09°].
* * *
$^{140}$removed: volume

$^{141}$removed: $v_r = i_r \, t_r$

$^{142}$removed: $t_r$ is the time interval

$^{143}$removed: times

$^{144}$removed: $LS_{3D}(x,y)$

$^{145}$removed: flow accumulation, representing the

$^{146}$removed: ,

$^{147}$removed: The empirical coefficients $m$ and $n$ for the upslope contributing area and the slope can range from 0.2 to 0.6 and 1.0 to 1.3 respectively with low values representing dominant sheet flow and high values representing dominant rill flow.

$^{148}$removed:

$$LS_{3D} = (m + 1.0) \, (a(x,y) \, a_0^{-1})^m \, (sin(\beta) \, \beta_0^{-1})^n$$

$^{149}$removed: (length-slope)

The empirical coefficients $m$ and $n$ for the upslope contributing area and the slope can range from 0.2 to 0.6 and 1.0 to 1.3 respectively with low values representing dominant sheet flow and high values representing dominant rill flow.

10 **2.3.4 [..[150] ]Detachment limited erosion rate**

[..[151] ]The erosion rate is a function of the event-based erosivity factor, [..[152] ]soil erodibility factor, [..[153] ]3D topographic factor, [..[154] ]landcover factor, and [..[155] ]prevention measures factor (Fig. 3[..[156] ]d):

$$E = R_e \ K \ LS_{3D} \ C \ P \tag{23}$$

where:

15     $E$ is [..[157] ]soil erosion rate (soil loss) $[\mathrm{kg\,m^{-2}\,min^{-1}}]$

    $R_e$ is the event-based erosivity factor $[\mathrm{MJ\,mm\,ha^{-1}\,hr^{-1}}]$

    $K$ is the soil erodibility factor $[\mathrm{ton\,ha\,hr\,ha^{-1}\,MJ^{-1}\,mm^{-1}}]$

    $LS_{3D}$ is the dimensionless topographic (length-slope) factor

    $C$ is the dimensionless [..[158] ]landcover factor

20     $P$ is the dimensionless prevention measures factor.

[..[159] ]The detachment limited erosion represented by RUSLE3D leads to the simulated change in elevation[..[160] ]:

$$\Delta z_s = D_c \rho_s^{-1} = E \ \rho_s^{-1} \tag{24}$$

which is combined with Eq. (3) for gravitational diffusion.

5 **2.4   Unit [..[161] ]Streampower Erosion Deposition (USPED)**

[..[162] ]USPED estimates net erosion-deposition as the divergence of sediment flow in transport capacity limited soil erosion [..[163] ]regime. The amount of soil detached is close to the amount of sediment that water flow can carry. As a transport
* * *
[150]removed: Sediment flow

[151]removed: Sediment flow

[152]removed: the

[153]removed: the

[154]removed: cover

[155]removed: the

[156]removed: f

[157]removed: sediment flow

[158]removed: land cover

[159]removed: For

[160]removed: $\Delta z$ is derived from equation **??** for landscape evolution in an detachment limited soil erosion regime and then equation 3for the settling of sediment particles due to

[161]removed: streampower erosion deposition model

[162]removed: The Unit Stream Power Erosion Deposition (USPED ) model

[163]removed: regimes. At transport capacity shallow flows of water are carrying as much sediment possible – more sediment is being detached than can be transported

capacity limited model USPED predicts erosion where transport capacity increases and deposition where transport capacity decreases. The influence of topography on [..[164] ]sediment flow is represented by a topographic sediment transport factor,

10   while the influence of soil and landcover are represented by factors adopted from USLE and RUSLE (Mitasova et al., 1996). [..[165] ]Sediment flow is estimated by computing the event-based erosivity factor ($R_e$) using Eq. 21, the slope and aspect of the topography, the flow accumulation with a multiple flow direction algorithm, the topographic sediment transport factor, [..[166] ]and sediment flow at transport capacity[..[167] ]. Net erosion-deposition is then computed as the divergence of [..[168] ]sediment flow.

15   **2.4.1**   Topographic sediment transport factor

[..[169] ]Using the unit stream power concept presented by Moore and Burch (1986), the 3D topographic factor (Eq. 22) for RUSLE3D is [..[170] ]modified to represent the topographic sediment transport factor ([..[171] ]$LS_T$) – the topographic component of overland flow at sediment transport capacity: [..[172] ]

$$LS_T = a^m (\sin\beta)^n \tag{25}$$

where:

[..[173] ]$LS_T$ is the topographic sediment transport factor

$a$ is the upslope contributing area per unit width [[..[174] ]m]

$\beta$ is the angle of the slope [°]

5   $m$ is an empirical coefficient

$n$ is an empirical coefficient.
* * *
[164] removed: erosion and deposition in USPED
[165] removed: Net erosion-deposition
[166] removed: the
[167] removed: , and
[168] removed: the
[169] removed: The
[170] removed: adapted
[171] removed: $LST$
[172] removed:

$$LST = a^m (\sin\beta)^n$$

[173] removed: $LST$
[174] removed: m

**2.4.2 Transport limited sediment flow and net erosion-deposition**

[..[175] ]Sediment flow at transport capacity is a function of the event-based rainfall factor, [..[176] ]soil erodibility factor, [..[177]

10  ]topographic component of overland flow, [..[178] ]landcover factor, and [..[179] ]prevention measures factor:

$$T = R_e \ K \ C \ P \ LST$$

[..[180] ]
[..[181] ][[..[182] ]]
[..[183] ][[..[184] ]]
15  [..[185] ][[..[186] ]]
[..[187] ]
[..[188] ]

$$T = R_e \ K \ C \ P \ LS_T \tag{26}$$

20  where:

$T$ is sediment flow at transport capacity $[\mathrm{kg\,m^{-1}\,s^{-1}}]$

$R_e$ is the event-based rainfall factor $[\mathrm{MJ\,mm\,ha^{-1}\,hr^{-1}}]$

$K$ is the soil erodibility factor $[\mathrm{ton\,ha\,hr\,ha^{-1}\,MJ^{-1}\,mm^{-1}}]$

$C$ is the dimensionless land cover factor

$P$ is the dimensionless prevention measures factor.

Net erosion-deposition [..[189] ]is estimated as the divergence of sediment flow, assuming that sediment flow is equal to sediment transport capacity:

$$d_s = \frac{\partial(T\,\cos\alpha)}{\partial x} + \frac{\partial(T\,\sin\alpha)}{\partial y}$$
* * *
[175] removed: The sediment
[176] removed: the
[177] removed: the
[178] removed: the
[179] removed: the
[180] removed: where:
[181] removed: $T$ is sediment flow at transport capacity
[182] removed: $\mathrm{kg\,m^{-1}\,s^{-1}}$
[183] removed: $R_e$ is the event-based rainfall factor
[184] removed: $\mathrm{MJ\,mm\,ha^{-1}\,hr^{-1}}$
[185] removed: $K$ is the soil erodibility factor
[186] removed: $\mathrm{ton\,ha\,hr\,ha^{-1}\,MJ^{-1}\,mm^{-1}}$
[187] removed: $C$ is the dimensionless land cover factor
[188] removed: $P$ is the dimensionless prevention measures factor.
[189] removed: at transport capacity

10   [..191 ][[..192 ]]

[..193 ][[..194 ]][..195 ]

$$d_s = \frac{\partial(T \cos\alpha)}{\partial x} + \frac{\partial(T \sin\alpha)}{\partial y} \tag{27}$$

where:

15   $d_s$ is net erosion-deposition $[\mathrm{kg\,m^{-2}\,s^{-1}}]$

$\alpha$ is the aspect of the topography (i.e. the direction of flow) $[°]$.

With USPED the simulated change in elevation [..196 ]$\Delta z_s = d_s$ is derived from equation 2 for landscape evolution and then equation 3 for [..197 ]gravitational diffusion.

20  **3   Case study**

[revised manuscript text omitted]

5   & 6c) – shows a deepening of the main channel by approximately 0.2 m and [..216 ]scours pits by approximately 1 m, while depositional ridges have formed and grown up to approximately 1 m [..217 ]high. The DoD also shows that 244.60 $m^3$ of sediment were deposited on the depositional fan between 2012 and 2016.
* * *
207 removed: (Figure 5a-c).
208 removed: 0.3
209 removed: by fusing
210 removed: with
211 removed: mannings
212 removed: rates
213 removed: Figure 5 d-f
214 removed: has
215 removed: between 2012 and 2016 (Figure 5d)
216 removed: the
217 removed: or more.

[Figure]

**Figure 5.** Morphological Change, Study Subwatershed, Patterson Branch[..[218] ], Fort Bragg, NC, USA. (a) Landcover in 2014, (b) landforms in 2012, (c) elevation difference between 2012-2016 [m], and (d) landforms in 2016.

[Figure]

**Figure 6.** Detailed Morphological Change, Drainage Area 1, Study Subwatershed, Patterson Branch, Fort Bragg, NC, USA. (a) Land-cover in 2014, (b) landforms in 2012, (c) elevation difference between 2012-2016 [m], and (d) landforms in 2016.

**3.2 Simulations**

We ran a sequence of r.sim.terrain simulations with design storms for the Patterson Branch [..²¹⁹ ]subwatershed study area to

10   [..²²⁰ ]demonstrate the capabilities of the RUSLE3D, [..²²¹ ]USPED, and SIMWE models (Table 2). [..²²² ]To analyze the results of the simulations, we compared net differences in elevation morphological features, and volumetric change. While r.sim.terrain can use rainfall records, we used design storms to demonstrate and test the basic capabilities of the model. Our design storms were based off the peak rainfall values in records from the State Climate Office of North Carolina. We used RUSLE3D to simulate landscape evolution in a dynamic, detachment capacity limited soil erosion regime for a

15   120 min [..²²³ ]design storm with 3 min intervals and a constant rainfall intensity of 50 mm hr$^{-1}$ [..²²⁴ ](Figure [..²²⁵ ]7). [..²²⁶ ][..²²⁷ ][..²²⁸ ]We used USPED to simulate landscape evolution in a dynamic, transport capacity limited soil erosion [..²²⁹ ]regime for a 120 min design storm with 3 min intervals and a constant rainfall intensity of 50 mm hr$^{-1}$ (Figure [..²³⁰ ]8). [..²³¹ ]We used SIMWE to simulate landscape evolution in a steady state, variable erosion-deposition soil erosion regime for a 120 min [..²³² ]design storm with a constant rainfall intensity of 50 mm hr$^{-1}$ [..²³³ ](Figure [..²³⁴ ]9). In all

5   of the simulations a sink filling algorithm – an optional parameter in r.sim.terrain – was used to reduce the effects of positive feedback loops that cause the over-development of scour pits.

The simulations were automated and run in parallel using Python scripts that are available in the software repository. The simulations can be reproduced using these scripts and the study area dataset by following the instructions in the Open Science Framework repository at https://osf.io/tf6yb/. The simulations were run in GRASS GIS 7.4 on a desktop computer with 64-bit

10   Ubuntu 16.04.4 LTS, 8 x 4.20 GHz Intel Core i7 7700K CPUs, and 32 GB RAM. Simulations using SIMWE are far more computationally intensive than RULSE3D or USPED, but support multi-threading when compiled with OpenMP. Dynamic simulations of RUSLE3D and USPED [..²³⁵ ]took 2 min 36 s and 3 min 14 s respectively to run on a single thread, while [..²³⁶ ]the steady state simulation for SIMWE took 44 min [..²³⁷ ]51 s running on 6 threads (Table 2).
* * *
²¹⁹removed: Creek

²²⁰removed: test dynamic and steady state flow regimes in the SIMWE,

²²¹removed: and USPED

²²²removed: RUSLE3D was used to simulate

²²³removed: events with rainfall intensities

²²⁴removed: for detachment capacity limited soil erosion regimes for both dynamic and steady state flow regimes using RUSLE3D

²²⁵removed: **??**a-c

²²⁶removed: USPED was used to simulate 120

²²⁷removed: events with rainfall intensities of

²²⁸removed: $^{-1}$ for

²²⁹removed: regimes for both dynamic and steady state flow regimes

²³⁰removed: **??**d-f

²³¹removed: SIMWE was used to simulate

²³²removed: events with rainfall intensities

²³³removed: for erosion-deposition and detachment limited soil erosion regimes in steady state flow regimes

²³⁴removed: **??**

²³⁵removed: each took

²³⁶removed: steady state simulations for SIMWE each took 84

²³⁷removed: 13

**Table 2.** Landscape evolution simulations

[revised manuscript text omitted]

[245]removed: simulations predicted more realistic patterns of landscape evolution (Figure ??). For transport limited and

[246]removed: regimes SIMWE simulated channel wideningand

[247]removed: (Figure ??c). For a detachment limited soil erosion regime SIMWE simulated major erosion driving the continued development of the gully network including the spread of rills and the evolution of the nascent branch into a full fledged channel

[248]removed: ??f). The detachment limited simulation also formed extensive ridges beside the gully channels (Figure ??f), continuing the development of channel-side ridges observed in the 2012 and 2016 landform maps (Figure 5e-f).

[249]removed: Given the presence of an active gully with ridges along its banks, this landscape is dominated by a detachment limited soil erosion regime. The detachment limited SIMWE simulation generated the morphological features – the deeply incised gully channels, scour pits, and ridges along the channels – characteristic of its erosion regime, realistically simulating landscape evolution at the scale of a subwatershed. The erosion-deposition and transport limited SIMWE simulations also generated the morphological processes and features that would be expected in these regimes – gradual aggradation and the formation of a depositional ridge along the thalweg of the channel.

[250]removed: While RUSLE3D and USPED produced less realistic patterns of landscape evolution than SIMWE, these models were much faster and still generated

[251]removed: at regional scales, i.e. for landscapes greater than 10

[252]removed: $^2$.

[Figure]

**Figure 7.** Dynamic simulation with RUSLE3D for a 120 min event with a rainfall intensity of 50 mm hr$^{-1}$ at 1 m resolution for the Study Subwatershed (a-b) and Drainage Area 1 (c-d) on Patterson Branch, Fort Bragg, NC. The (a) flow accumulation and (b) erosion [kg m$^{-2}$ s$^{-1}$] for the Study Subwatershed in the final 3 min timestep. The (c) net difference [m] and (d) landforms for Drainage Area 1.

[Figure]

a.

b.

c.

d.

**Figure 8.** Dynamic [..²⁵³ ]simulation with USPED [..²⁵⁴ ]for a 120 min event with a rainfall intensity of 50 mm hr⁻¹ at 1 m resolution for the Study Subwatershed (a-b) and Drainage Area 1 (c-d) on Patterson Branch, Fort Bragg, NC. The (a) flow accumulation and (b) erosion-deposition $[\mathrm{kg\,m^{-2}\,s^{-1}}]$ for the Study Subwatershed in the final 3 min timestep. The (c) net difference $[\mathrm{m}]$ and (d) landforms for Drainage Area 1.

[Figure]

**Figure 9.** Steady state SIMWE simulations for a 120 min [..²⁵⁵ ]event with a rainfall [..²⁵⁶ ]intensity of 50 mm hr⁻¹ at 1 m resolution for the Study Subwatershed (a-b) and [..²⁵⁷ ]Drainage Area 1 (c-d) on Patterson Branch, Fort Bragg, NC. The (a) depth of unsteady flow [m] and (b) erosion-deposition [kg m⁻² s⁻¹] for the Study Subwatershed. The (c) net difference [m] and (d) landforms for Drainage Area 1.

10 solution of the sediment transport equation, the accuracy, detail, and smoothness of the results depend on the number of random walkers. While a large number of random walkers will reduce the numerical error in the path sampling solution, it will also greatly increase computation time. A customized compilation of GRASS GIS is needed to run SIMWE with more than 7 million random walkers. This limits the size of rasters that can be easily processed with SIMWE, while RUSLE3D and USPED are much faster, computationally efficient, and can easily be run on much larger rasters.

15    In the future we plan to assess this model by comparing simulations against a monthly timeseries of submeter resolution surveys by unmanned aerial systems and terrestrial lidar. We also plan to develop a case study demonstrating how the model can be used as a planning tool for landscape restoration. Planned enhancements to the model include modeling subsurface flows, accounting for bedrock, and a reverse landscape evolution mode for backward modeling.

**5  Conclusions**

20 The short-term landscape evolution model r.sim.terrain can [..[258] ]simulate the development of gullies, rills, and hillslopes by overland water erosion for a range of hydrologic and soil erosion regimes. The model is novel for simulating landscape evolution based on unsteady flows. The landscape evolution model was tested with a series of simulations for different hydrologic and soil erosion regimes for a highly eroded sub-watershed on Fort Bragg with an active gully. For each regime it generated the morphological processes and features expected. The physics-based SIMWE model [..[259] ]simulated morphological pro-
25 cesses [..[260] ]for a variable erosion-deposition [..[261] ]regime such as gradual aggradation, channel widening, scouring, the development of depositional ridges along the thalweg, and the growth of the depositional fan. The empirical RUSLE3D [..[262] ]model simulated channel incision in a detachment limited soil erosion [..[263] ]regime, while the semi-empirical USPED model simulated channel widening and filling [..[264] ]in a transport limited regime. Since r.sim.terrain is a GIS-based model that [..[265] ]simulates fine-scale morphological processes and features, [..[266] ]it can easily and effectively be used in conjunction
5 with other GIS-based tools for geomorphological research, land management and conservation, erosion control, and landscape restoration.

*Code and data availability.* As a work of open science this study is reproducible, repeatable, and recomputable. Since the data, model, GIS, dependencies are all free and open source, the study can easily be reproduced. The landscape evolution model has been implemented
* * *
[258]removed: realistically

[259]removed: realistically simulated short-term topographic change for steady state hydrologic regimes at sub-watershed to watershed scales. For detachment limited soil erosion regimes it

[260]removed: including channel incision, channel widening, and the development of knickzones, rills, and scour pits. For transport limited and

[261]removed: regimes, it simulated processes such as channel aggradation, scouring, and

[262]removed: and USPED models approximated short-term topographic change at watershed to regional scales. For

[263]removed: regimes RUSLE3D simulated channel incision, while for transport limited regimes USPED

[264]removed: . Since it

[265]removed: realistically

[266]removed: r.sim.terrain

[revised manuscript text omitted]